# Inhibition of the H3K27 demethylase UTX enhances the epigenetic silencing of HIV proviruses and induces HIV-1 DNA hypermethylation but fails to permanently block HIV reactivation

**Kien Nguyen**[1]*, **Curtis Dobrowolski**[2], **Meenakshi Shukla**[1], **Won-Kyung Cho**[3], **Benjamin Luttge**[1], **Jonathan Karn**[1]

**1** Department of Molecular Biology and Microbiology, Case Western Reserve University Medical School, Cleveland, Ohio, United States of America, **2** Wallace H. Coulter Department of Biomedical Engineering, Georgia Institute of Technology and Emory School of Medicine, Atlanta, Georgia, United States of America, **3** Korean Medicine (KM)-Application Center, Korea Institute of Oriental Medicine (KIOM), Dong-gu, Daegu, Republic of Korea

* Kien.Nguyen@case.edu

**Data Availability Statement:** All relevant data are within the manuscript and its Supporting information files.

## Abstract

One strategy for a functional cure of HIV-1 is "block and lock", which seeks to permanently suppress the rebound of quiescent HIV-1 by epigenetic silencing. For the bivalent promoter in the HIV LTR, both histone 3 lysine 27 tri-methylation (H3K27me3) and DNA methylation are associated with viral suppression, while H3K4 tri-methylation (H3K4me3) is correlated with viral expression. However, H3K27me3 is readily reversed upon activation of T-cells through the T-cell receptor. In an attempt to suppress latent HIV-1 in a stable fashion, we knocked down the expression or inhibited the activity of UTX/KDM6A, the major H3K27 demethylase, and investigated its impact on latent HIV-1 reactivation in T cells. Inhibition of UTX dramatically enhanced H3K27me3 levels at the HIV LTR and was associated with increased DNA methylation. In latently infected cells from patients, GSK-J4, which is a potent dual inhibitor of the H3K27me3/me2-demethylases JMJD3/KDM6B and UTX/KDM6A, effectively suppressed the reactivation of latent HIV-1 and also induced DNA methylation at specific sites in the 5'LTR of latent HIV-1 by the enhanced recruitment of DNMT3A to HIV-1. Nonetheless, suppression of HIV-1 through epigenetic silencing required the continued treatment with GSK-J4 and was rapidly reversed after removal of the drug. DNA methylation was also rapidly lost after removal of drug, suggesting active and rapid DNA-demethylation of the HIV LTR. Thus, induction of epigenetic silencing by histone and DNA methylation appears to be insufficient to permanently silence HIV-1 proviral transcription.

## Author summary

The "block and lock" strategy for a functional HIV-1 cure is based on the premise that permanent inactivation of the HIV-1 can be achieved by epigenetic silencing of the

**Funding:** This work was supported by US Public Health Service grants R01 MH110360, R01 DA043159, R01 AI148083 and amfAR grant 109347-59-RGRL to JK. The funders had no role in study design, data collection and analysis, decision to publish, or preparation of the manuscript.

**Competing interests:** The authors have declared that no competing interests exist.

proviral DNA. For cellular genes, long-term silencing is achieved during cell differentiation by the induction of specific epigenetic modifications involving histone and DNA methylation. During HIV-1 silencing, histone methylation and DNA methylation are observed, but both sets of modifications can be reversed upon activation of T-cells through the T-cell receptor or potent latency reversing agents. In an attempt to enhance silencing of HIV-1 transcription, we used an inhibitor of H3K27 demethylases to increase H3K27 methylation. This in turn led to enhanced DNA methylation of HIV-1. Unfortunately, although the treatment effectively silenced HIV-1 and prevented viral reactivation, the silencing effects were short-lived and quickly reversed after removal of the drug.

## Introduction

Despite effective long-term suppression of viral loads by combination antiretroviral therapy (cART), latent HIV-1 may quickly rebound upon cessation of treatment [1, 2]. There are two major strategies for eliminating latent proviruses or preventing the rebound of latent proviruses. The first, often referred to as "kick and kill" [3], is based on the concept that latent HIV-1, once pharmacologically reactivated by latency reversing agents (LRAs), could subsequently be eliminated by cytotoxicity and/or the enhanced immunological surveillance [4, 5]. Unfortunately, "kick and kill" faces formidable challenges in achieving reactivation of the majority of latent provirus [6–10] and difficulties in inducing HIV-1 specific immune-mediated clearance *in vivo* [11–13]. An alternative approach, referred to as "block and lock", is designed to permanently suppress the rebound of quiescent HIV-1 by inducing epigenetic silencing [14–16]. In the best documented example of a "block and lock" strategy, the Valente laboratory has reported that didehydro-cortistatin A (dCA), an inhibitor of Tat-mediated HIV-1 transcription, can enhance epigenetic silencing of the HIV-1 promoter, leading to long-term inactivation of the provirus [14, 17–19]. Regardless of the chosen strategy, a more comprehensive understanding of the mechanism involved in HIV-1 latency is urgently required when designing effective HIV-1 cure strategies.

Multiple complementary molecular mechanisms contribute to the development of HIV-1 latency in T-cell populations, including sequestration of key transcription initiation and elongation factors [20]. In addition, numerous epigenetic modifications of histones on the HIV-1 LTR, including H3K27 and H3K9 methylation, serve to silence HIV-1 proviruses [21–28].

H3K27 trimethylation (H3K27me3) provides one of the essential silencing mechanisms of HIV-1 in CD4$^+$ T cells [22, 23, 29–32]. EZH2, the enzymatic moiety of the polycomb repressive complex 2, is the sole H3K27 methyltransferase (H3K27MT) in mammalian cells. Opposing the H3K27MT is UTX/KDM6A, one of two specific H3K27 demethylases, containing highly homologous Jumonji C domains [33, 34], that selectively demethylates H3K27me3 and H3K27me2. UTX is essential for the activation of transcriptionally repressed genes containing bivalent promoters [35, 36]. UTX is a component of the UTX-MLL2/3 complexes, which maintain the transcriptionally positive epigenetic mark H3K4me3.

In addition to histone methylation, promoters are often permanently silenced by DNA methylation, which occurs when cytosine residues are methylated by DNA methyltransferases (DNMTs) to create 5-methylcytosine (5mC) residues [37]. The majority of DNA methylation occurs at cytosine in the CpG islands [38]. Methylated DNA itself suppresses the transcription of cellular genes by eliminating the binding of transcription activators [37]. In addition, methylation of DNA also promotes the binding of methylated-DNA binding proteins which usually

function as transcription co-repressors [37, 39, 40]. DNA methylation of CpG islands in the promoter is usually associated with long-term transcription repression [41].

DNA methylation at two CpG islands flanking the transcription start site in the HIV-1 5' LTR has been associated with HIV-1 silencing [42, 43]. However, the latent viral reservoir of cART-treated aviremic patients displays very low or no DNA methylation, suggesting that direct DNA modification is not a major control mechanism for latency [44, 45].

We reasoned that since H3K27me3 is essential for HIV-1 latency in T cells [22, 23] enhancement of H3K27me3 by inhibition of H3K27 demethylases, particularly UTX, could lead to permanent silencing of the provirus. Blocking UTX was able to prevent proviral reactivation and led to increases in both H3K27me3 and DNA methylation at specific sites in the 5' LTR. Unfortunately, this did not result in irreversible suppression of latent HIV-1 in T cells, suggesting that epigenetic silencing is a highly dynamic process that cannot by itself lead to a permanent block to HIV-1 rebound.

## Results

### Depletion of UTX expression blocks HIV-1 reactivation

UTX/KDM6A is a specific H3K27 demethylase that selectively targets H3K27me3 and H3K27me2. Since H3K27me3 formation by EZH2 is a powerful repressive mechanism to silence HIV-1 [22, 23], we reasoned that knockdown of UTX could be used to enhance HIV-1 transcriptional silencing and perhaps establish a permanently silenced provirus. As shown in Fig 1, we initially performed knockdown studies in the latently infected Jurkat E4 cell line [22, 23, 46, 47] using two different shRNAs designed to target UTX (i.e. shUTX(1) & shUTX(2)) along with a scrambled shRNA control. The shRNAs were introduced into the E4 cells using a lentiviral expression vector and single cells were then sorted into wells to derive clones.

As shown in Fig 1A, after reactivation of the latent proviruses by TNF-α in E4 cells, up to 76% GFP+ cells were obtained in the cells expressing the scrambled shRNA control, while we observed less than 30% viral reactivation in two representative clones expressing two different shRNAs to UTX (shUTX-cl1; shUTX-cl2). Similar blocks to HIV-1 reactivation by SAHA were also observed (Fig 1B). We also confirmed the inhibitory effect of UTX knockdown on HIV-1 using two other latently infected Jurkat cell lines: 2D10 cells, which were infected with Nef- H13L Tat HIV-1 and 3C9 cells, harboring Nef+ H13L Tat HIV-1 (Fig 1B) [22, 23, 46, 47].

When working with a population of latently infected cells carrying HIV, the positive feedback mechanism mediated by Tat induction results in high expression levels, resulting in a distinct population of activated cells [22, 23, 46]. Different degrees of activation therefore primarily result in shifts in the fraction of cells that express HIV markers (including GFP), in contrast to most promoters where the intensity of gene expression may shift over a large dynamic range. Therefore, in Fig 1 and the rest of this paper, when analyzing the flow cytometry data, we express HIV reactivation data in terms of changes in the % of HIV (GFP) positive cells. The FACs data can also be expressed as a fold change in the arithmetic mean fluorescence intensity (MFI) of the whole population of cells. To demonstrate that the results are similar with both methods of analysis, we plotted our data from the HIV-1 reactivation experiment performed on UTX knocked down E4 clones (Fig 1B) in S1 Fig. Both approaches clearly document the highly statistically significant inhibitory effect of UTX knock down on HIV-1 reactivation (S1 Fig). However, the nominal background in the MFI data appears higher than in the % GFP+ cell data because of the contributions of the small population of cells that are already activated.

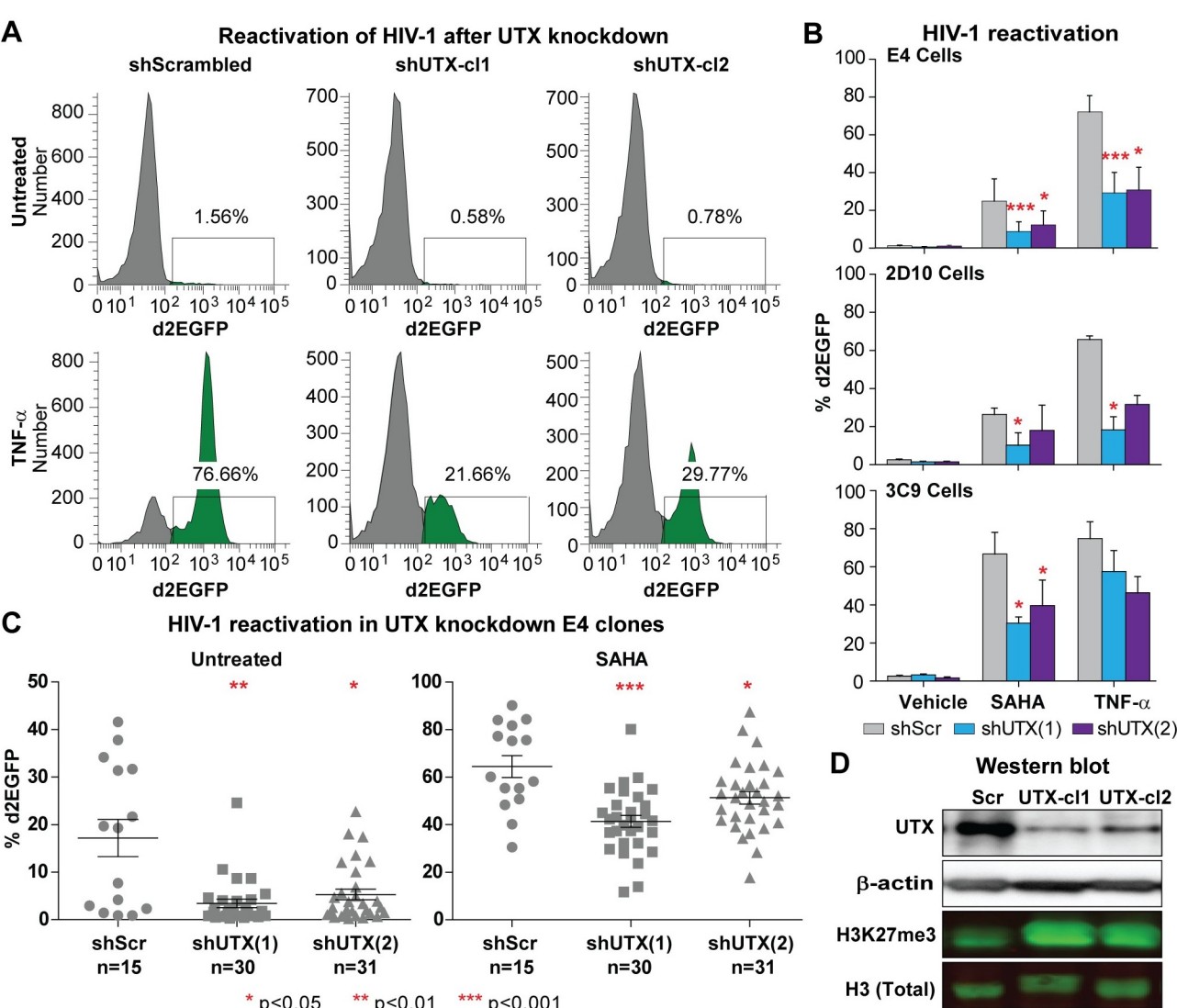

**Fig 1. Inhibition of HIV-1 reactivation in UTX knocked down E4 cells.** (A) Representative FACS measuring d2EGFP expression in two UTX knocked down clones from E4 cells stimulated with TNF-α (10 ng/ml) overnight. (B) Quantification of HIV-1 reactivation in two clones from UTX knocked down E4 cells, UTX knocked down 2D10, and UTX knocked down 3C9 cells stimulated with indicated conditions. 3C9 cells harbor Nef+, Wt Tat HIV, while 2D10 cells are infected with H13L mutant Tat HIV-1. Note that H13L Tat is a functional Tat, as it has been shown that we can easily reactivate HIV-1 in 2D10 cells by many different stimuli. Error bars: SEM of 3 separate experiments. (C) d2EGFP expression in UTX knocked down E4 clones left untreated or stimulated with SAHA (2 μM) overnight. Kruskal-Wallis test was used for statistical calculation. (D) Western blot measuring the levels of UTX and H3K27me3 in E4 cells expressing scramble or UTX shRNAs.

To verify that the inhibition on HIV-1 reactivation was not clonally dependent, multiple clones of E4 cells transduced with either the scrambled or UTX shRNAs were reactivated using 2 μM of SAHA overnight (Fig 1C). In this data set, each point represents an independent clone, which show a range of basal expression levels. Expression of both UTX shRNA sequences significantly inhibited the basal transcription of provirus, resulting in a large fraction of clones that produced little detectable GFP in the absence of activation. Nonetheless, these proviruses can be reactivated following treatment with 2 μM SAHA. For both shUTX sequences, the average fraction of reactivated cells was reduced compared to the scrambled

shRNA controls, although the shUTX(2) sequence was somewhat less efficient than shUTX(1) in blocking HIV-1 reactivation.

The relative potency of the two shRNAs was confirmed by Western blots using a specific UTX antibody (Fig 1D). UTX levels were depleted by more than 70% using both UTX shRNAs, with shUTX(1) showing somewhat greater activity than shUTX(2), consistent with their relative activity in blocking HIV-1 transcription. As expected, UTX depletion also resulted in elevation of cellular H3K27me3 levels, and as expected, H3K27me3 levels were highest in the UTX-cl1 treated cells. In summary, we were able to demonstrate in three latently infected Jurkat T-cell lines and in multiple clones derived from two unique shRNAs that UTX depletion enhances HIV-1 silencing and blocks reactivation of HIV-1 transcription.

## Enhanced recruitment of UTX to the HIV-1 LTR during proviral reactivation

UTX activates HIV-1 provirus by converting the LTR chromatin structure to a more transcriptionally active state by up-regulation of H3K4 methylation and down-regulation of H3K27 methylation [48]. We confirmed and expanded these observations using chromatin immuno-precipitation (ChIP) assays [22, 23, 47] to measure the accumulation of RNAP II, UTX, EZH2, and H3K27me3 at the LTR of reactivated proviruses (Fig 2).

After stimulation of latent proviruses with 10 ng/ml TNF-α for an hour, we observed a more than 10-fold increase in RNAP II recruitment to the Nuc-0, promoter, Nuc-1, and Nuc-2 regions of HIV-1 LTR (Fig 2A), indicating the initiation of viral transcription. The increase in RNAP II was correlated with recruitment of UTX. Consistent with our previous reports documenting EZH2 repression of HIV-1 transcription in T cells [22, 23], proviral reactivation resulted in corresponding reductions in the levels of EZH2 and H3K27me3.

ChIP assays were also performed on an E4 clone expressing UTX(1) shRNA and a control clone expressing the scrambled shRNA (Fig 2B). Depletion of UTX resulted in a reduction of RNAP II occupancy throughout the LTR. The expected removal of UTX throughout the HIV-1 provirus was demonstrated as an additional control for shRNA potency. In addition, we also detected an increase of H3K27me3 and decrease of H3K4me3 at HIV-1 LTR. Together, our data demonstrates that UTX is recruited to HIV-1 LTR and facilitates the initiation of HIV-1 transcription.

We routinely use an IgG control antibody in our ChIP assays. To simplify the graphs for the data shown in Fig 2 and throughout this paper, the signals generated by IgG antibody were subtracted from the signals obtained with the specific antibodies. The means and standard deviations of ChIP signals from IgG antibody and specific antibodies have also been presented in S2 Fig.

Because the HIV LTR is duplicated at both ends of the provirus, a limitation of all ChIP assays performed on the LTR is that it is difficult to distinguish between proteins accumulating exclusively on the 5'LTR [47]. In these experiments, we used primers selective for the 5' LTR for the Nuc-1 and Nuc-2 regions, but we were unable to selectively prime on Nuc-0 and the promoter in these experiments. However, our previous ChIP experiments showed that RNAP II accumulates virtually entirely at the 5' LTR along with other transcription elongation factors such as P-TEFb and NELF [47] and it seems extremely likely that UTX and the rest of histone methylation machinery show a similar distribution. In any case, the accumulation of UTX at Nuc-1 and Nuc-2 downstream of the 5' LTR is unambiguous.

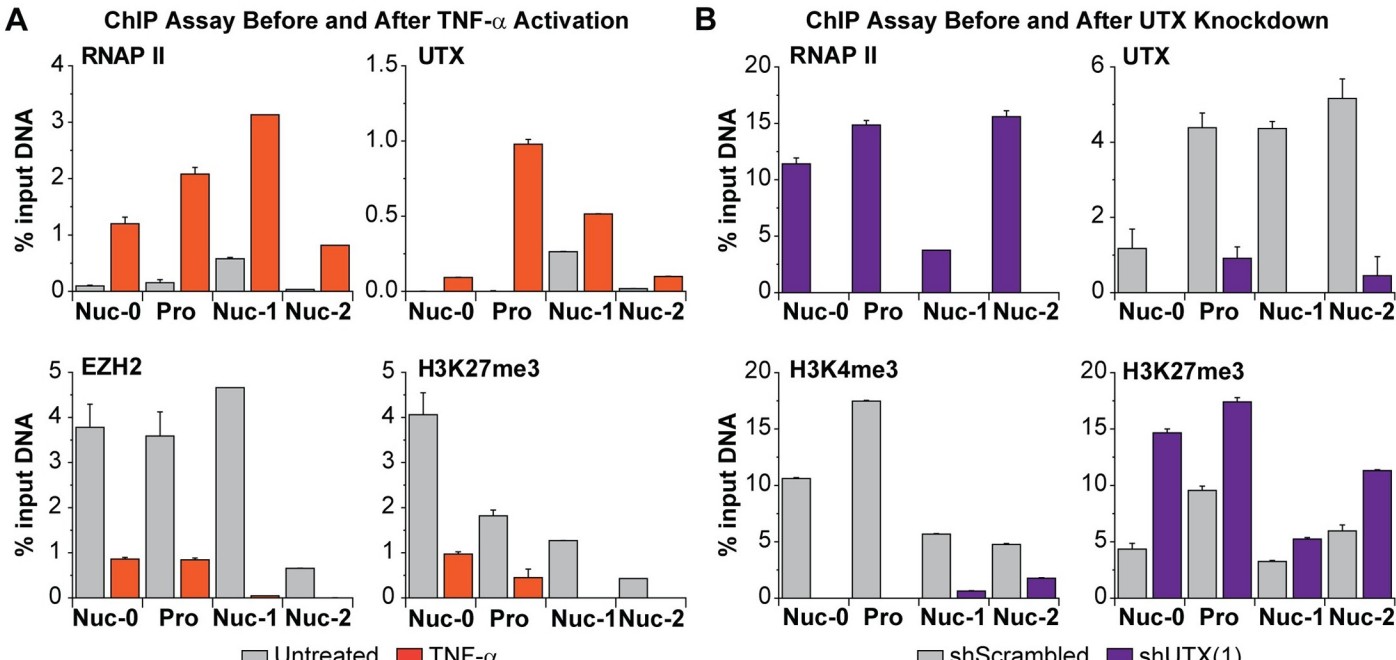

**Fig 2. UTX functions as a transcription activator of HIV-1 transcription.** (A) ChIP assays measuring the enrichments of RNAP II, UTX, EZH2, and H3K27me3 at the Nuc-0, HIV-1 promoter, Nuc-1, and Nuc-2 regions of HIV-1 before or after TNF-α reactivation. Latent proviruses in E4 cells were reactivated by TNF-α (10 ng/ml) for an hour. Cells were treated with ethylene glycol bis(succinimidyl succinate) (EGS) (1.5 mM) for 30 minutes and subsequently crosslinked with formaldehyde (1%) for additional 10 minutes. (B) ChIP assays measuring the enrichment of RNAPII, UTX, H3K4me3, and H3K27me3 along HIV-1 genome in one UTX knocked down clone. Error bars: SEM of 3 separate quantitative real-time PCRs. Note that Nuc-0 and HIV-1 promoter primers amplify both 5'LTR and 3'LTR of HIV-1; Nuc-1 and Nuc-2 primers specifically bind to 5'LTR.

## The HIV-1 LTR is a bivalent promoter and requires UTX for reactivation

Bivalent promoters, which include the HIV-1 LTR [48], characteristically contain both the activating H3K4me3 and repressive H3K27me3 marks at the same nucleosomes and are usually maintained at a repressed, but rapidly inducible, transcriptional state [49, 50]. The important role of UTX demethylase activity for the resolution and activation of genes containing bivalent promoters has been documented [35, 36].

To directly evaluate whether the UTX demethylase activity participates in HIV-1 transcriptional reactivation, we inhibited H3K27me3 using GSK-J4, a specific antagonist for both UTX and JMJD3 demethylase activity [51–53]. E4 and 3C9 cells were pretreated for 24 hours with 5 or 10 μM of GSK-J4, then further stimulated with 2 μM SAHA or 10 ng/ml TNF-α overnight. As shown in Fig 3A, a dose-dependent inhibition of HIV-1 reactivation was observed in both E4 and 3C9 cells when cells were treated with GSK-J4. This was seen in all tested concentrations of SAHA and TNF-α. With 10 μM of GSK-J4, a maximal reduction of more than 50% in the reactivation levels of provirus was obtained in both cell lines. Representative histograms showing the FACS analysis of HIV-1 reactivation are shown in S3 Fig. Only viable cells were chosen for FACs analysis. As shown by the $EC_{50}$ and $CC_{50}$ data for GSK-J4 on E4, 3C9 cells treated for 24 or 48 hours (S4 Fig), there is a buildup of K27me3+ over time and there is some toxicity starting at 10 μM of GSK-J4 for the transformed cell lines which is not evident in primary Th17 cells.

ChIP performed on E4 cells to measure the density of RNAP II, H3K4me3, and H3K27me3 at the LTR of silenced provirus confirms the bivalent nature of the HIV-1 promoter and the

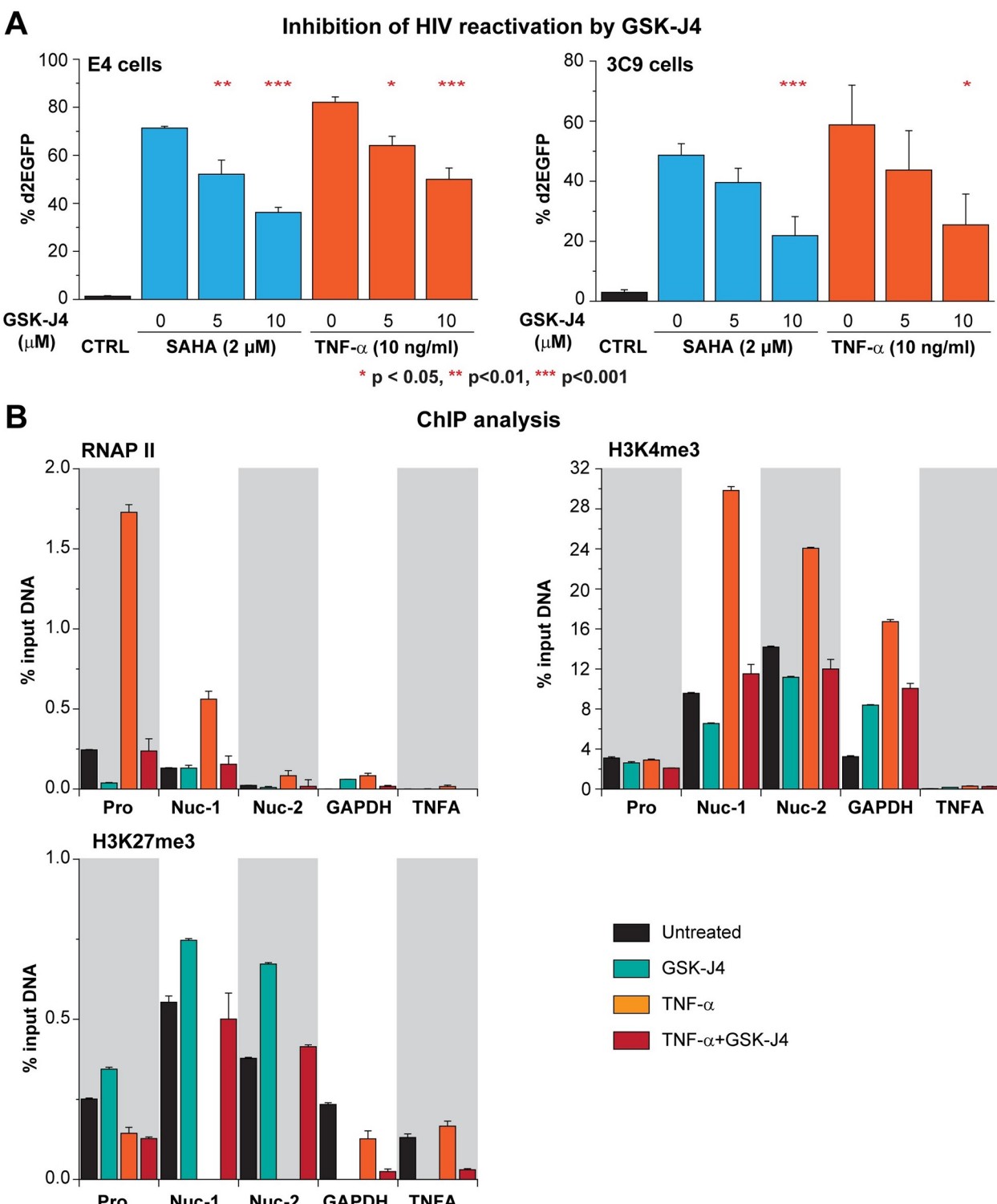

**Fig 3. Inhibition of UTX demethylase activity by GSK-J4 prevents HIV-1 reactivation in Jurkat T cells.** (A) Quantification of d2EGFP expression in E4 and 3C9 cells pretreated with increasing concentrations of GSK-J4 for 24 hours and stimulated with SAHA (2 μM) or TNF-α (10 ng/ml) overnight. Error bars: SEM of 5 separate experiments. One-way ANOVA, P <0.05, Bonferroni posttests * p < 0.05, ** p<0.01, *** p<0.001, n = 5. (B) ChIP assays measuring the enrichments of RNAP II, H3K4me3, and H3K27me3 at the 5'LTR of HIV-1 when latent proviruses were left unstimulated or reactivated for an hour by TNF-α (10 ng/ml) with or without the presence of GSK-J4 (5 μM). E4 cells were pretreated for 24 hours with GSK-J4 (5 μM) then further stimulated with TNF-α (10 ng/ml) for an hour. Error bars: SEM of 3 separate quantitative real-time PCRs.

ability of GSK-J4 to block proviral reactivation (Fig 3B). In untreated cells, RNAP II accumulates near the promoter and Nuc-1 (Fig 3B) [47]. Nuc-1 and Nuc-2 are simultaneously occupied by both methylated histones (Fig 3B) [48]. Treatment of HIV-1 with 5 μM GSK-J4 for 24 hrs resulted in a modest increase of H3K27me3 and decrease of H3K4me3 at HIV-1 LTR, similar to what was observed in UTX knocked-down cells. ChIP signals from IgG antibody and specific antibodies have also been presented in S5 Fig.

Upon TNF-α stimulation in the absence of GSK-J4, there was enhanced recruitment of RNAP II to HIV-1 promoter region (-116 to +4 position relative to transcription start site (TSS)), a direct measurement of transcriptional reactivation of HIV-1 [47]. Concomitantly, H3K4me3 levels at the Nuc-1 and Nuc-2 regions were substantially increased, and as expected, H3K27me3 was completely removed at the same regions [22, 23]. By contrast, when the provirus was reactivated by TNF-α in the presence of GSK-J4, the levels of H3K4me3 and H3K27me3 were fully restored, while RNAP II levels were reduced (Fig 3B). The changes in H3K4me3 and H3K27me3 levels are HIV-1 specific and not detected at the control cellular genes GAPDH and TNFA, which are not bivalent genes. These observations strongly support the conclusion that the H3K27 demethylase activity of UTX is essential for the reactivation of the silent provirus' bivalent LTR.

## Simultaneous demethylation of H3K27me3 and methylation of H3K4 are required for HIV-1 reactivation

The bivalency of the HIV-1 LTR and the association of UTX with the H3K4 methyltransferase MLL2/MLL3 complex strongly suggested that H3K4me3 is crucial for HIV-1 reactivation. To test this hypothesis, we ectopically expressed HA-tagged histone H3.3 containing either a wild type sequence, the single mutants K4M and K27M, or a double mutant K4M-K27M and investigated their impacts on HIV-1 reactivation. We chose H3.3 because H3.3 is associated with PRC2-dependent H3K27me3 deposition and gene expression regulation [54–56]. As reported in our previous papers, HIV-1 is Polycomb repressive complex 2 dependent [22]. Lysine to methionine (K to M) mutations of H3.3 have dominant negative effects on the global lysine methylation of histone H3 at different residues [57].

A substantial reduction in the cellular levels of H3K27me3 or H3K4me2-3 was observed in cells expressing K27M or K4M mutants of H3.3 (Fig 4A). As expected, the K4M and K27M were not substrates for histone methylation (Fig 4A). Compared to the wild type H3.3, the expression of the H3.3 K27M mutant resulted in spontaneous reactivation of HIV-1 and significantly sensitized the provirus to reactivation induced by 2 μM SAHA or 10 ng/ml TNF-α. By contrast, the K4M mutant had no significant effect on HIV-1 reactivation (Fig 4B). The double mutant K4M-K27M abolished the enhanced reactivation induced by the K27M single mutant and restored the phenotype of wild type H3.3 (Fig 4B). In control experiments, the incorporation of the wild type histone variant H3.3 into HIV-1 chromatin did not result in any enhancement of viral reactivation versus mock control cells (S6 Fig). The statistical significance of all of these results was based on comparisons with the H3.3 K27M variant to emphasize the necessity of H3K27me3 depletion on HIV-1 reactivation and the indispensable nature of H3K4me3 on HIV-1 reactivation mediated by H3K27me3 depletion. Detailed statistical analyses with different pairs of treatments are included in S1 Table.

We conclude that maintenance of H3K4me3 is required for the reactivation of HIV-1 transcription mediated by the demethylation of H3K27me3. This strongly indicates that UTX plays a dual role in HIV-1 reactivation: removal of H3K27me3 and recruitment of MLL2/3 complexes to deposit H3K4me3.

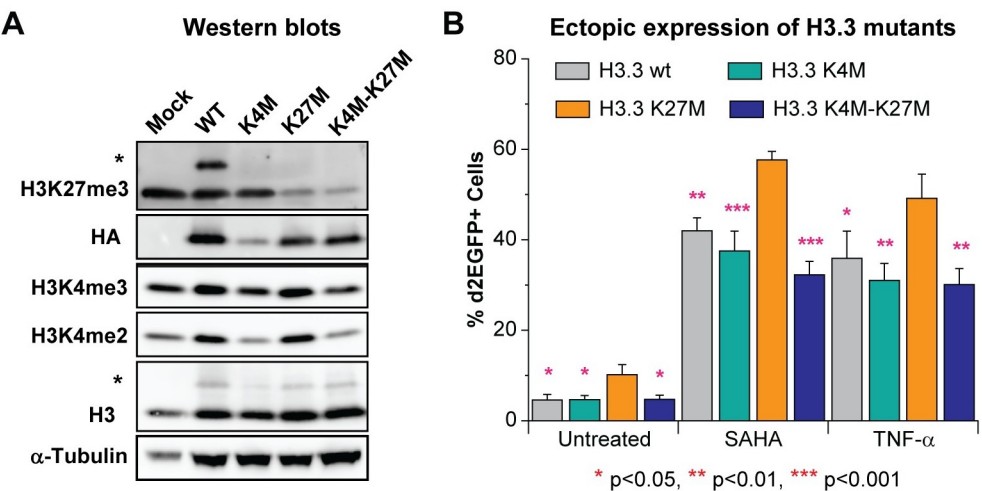

**Fig 4. H3K4 methylation is crucial for HIV-1 reactivation mediated by H3K27 removal.** (A) Western blot measuring the indicated histone H3 methylation levels of cells expressing HA tagged wild type (wt) H3.3 or indicated mutants of H3.3. The upper bands (marked with *) from blots with antibodies against H3K27me3 and H3 are from the H3.3-HA variants. (B) Quantification of HIV-1 reactivation in E4 cells expressing wt H3.3 or indicated H3.3 mutants. Presented statistical significance was based on comparisons with H3.3 K27M variant. One-way ANOVA, n = 5 $p < 0.005$, Bonferroni posttests, * $p < 0.05$, ** $p < 0.01$, *** $p < 0.001$. Error bars: SEM of 5 different replications.

## Inhibition of UTX induces DNA methylation of the HIV-1 LTR

DNA methylation is generally a more permanent epigenetic silencing mechanism than histone methylation alone. Often during development, a decrease of H3K4 methylation results in a corresponding increase in DNA methylation and permanent gene silencing [58–62]. Since depletion of UTX caused a decrease in H3K4me3 levels at HIV-1 LTR, we next tested whether UTX depletion also induced DNA methylation of HIV-1.

The HIV-1 genome carries a series of CpG islands that are subject to DNA methylation (Fig 5A). Although the functional role of DNA methylation in HIV-1 latency is controversial [44, 63–65], one commonly used approach for measuring the impact of DNA methylation on gene expression is to use 5'-Azacytidine (5-AZC), an analogue of cytosine that can be incorporated into DNA during DNA replication, to inhibit DNA methyltransferases (DNMTs). Binding of DNMTs to 5-AZC results in "covalent trapping" of DNMTs resulting in degradation of the enzymes [66]. An increase in gene expression by 5-AZC therefore implies removal of a repressive DNA methylation mark.

As shown in Fig 5B, E4 cells expressing either scrambled or UTX shRNAs were treated for 72 hours with 5-AZC (1 μM), then left untreated or stimulated further overnight either through the T-cell receptor (TCR) (using as combination of α-CD3 (125 ng/ml) and α-CD28 (500 ng/ml)), treatment with 500 nM of SAHA, or activation with 1 ng/ml of TNF-α. HIV-1 transcription was measured by monitoring d2EGFP expression levels using flow cytometry. Compared to the scrambled shRNA-treated control cells, we observed greater elevations of HIV-1 expression in the shUTX knock down cells after 5-AZC treatment (Fig 5B). When treated with 5-AZC, cells expressing scrambled shRNA showed a less than a 1.5-fold increase in HIV-1 expression while the UTX shRNA treated cells showed a 3 to 6-fold increase. Reactivation of the proviruses following treatment of 5-AZC pretreated cells through the TCR, SAHA, or TNF-α each enhanced reactivation of HIV-1 in shUTX knock down cells compared to the control cells.

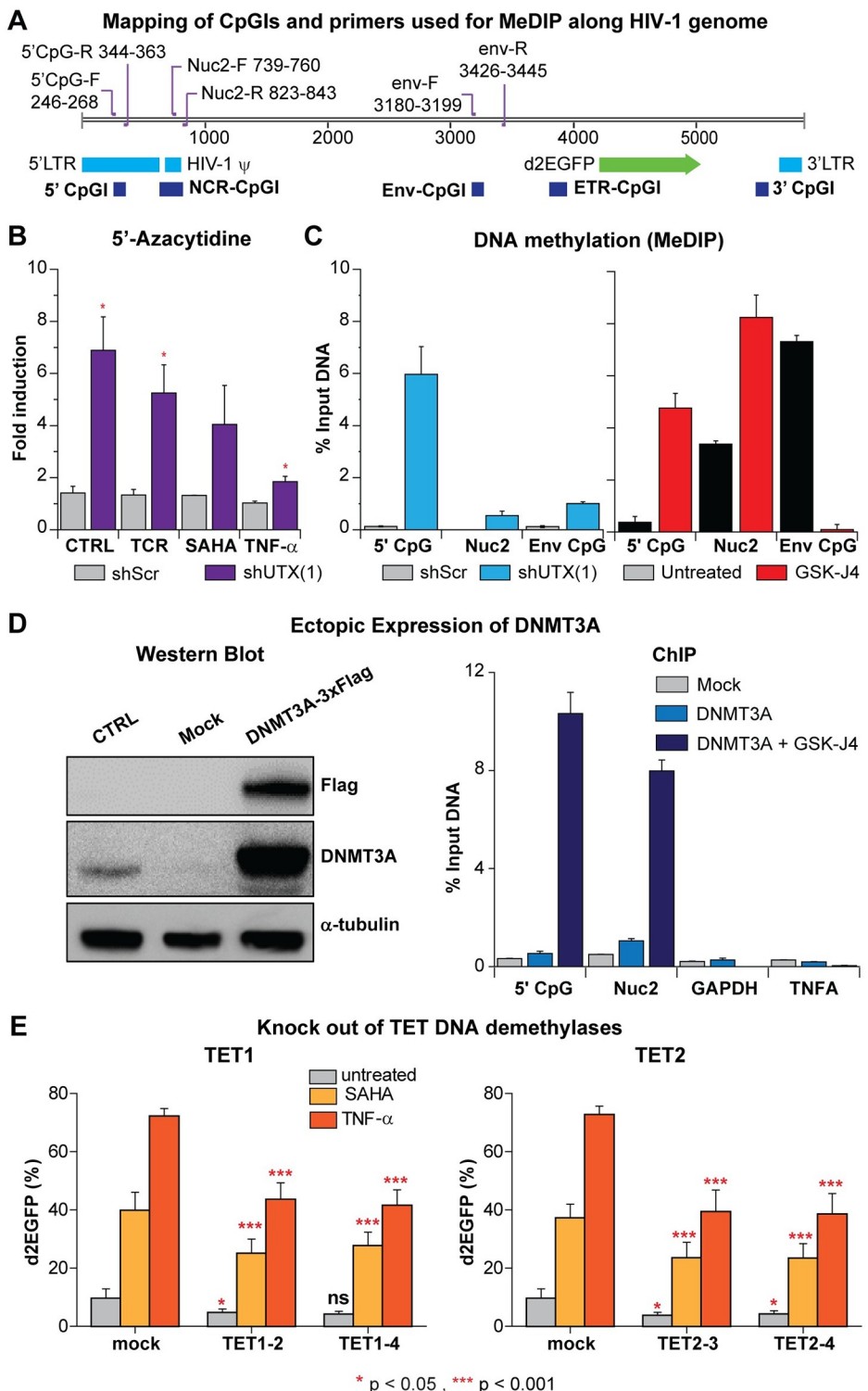

**Fig 5. Depletion of UTX elevates DNA methylation levels of latent HIV-1 in Jurkat T cells due to enhanced recruitment of DNMT3A to HIV-1 5'LTR.** (A) The map indicates the positions of CpG islands and primer binding sites along HIV-1 genome. (B) HIV-1 reactivation in UTX knocked down cells treated with 5-AZC. E4 cells were pretreated with 1 μM 5-AZC for 72 hours then left untreated or treated further with a combination of anti-CD3 (125 ng/ml) and anti-CD28 (500 ng/ml), 500 nM of SAHA, or 1 ng/ml of TNF-α overnight. HIV-1 reactivation in the cells was measured by FACS. Error bars: SEM of at least 3 separate replicates. T-test, n = 3, * p < 0.05. (C) MeDIP assay

measuring the levels of methylated cytosine of HIV-1 in UTX knock down E4 cells or E4 cells treated with 5 µM of GSK-J4 for 48 hours. Error bars: SEM of 3 separate quantitative real-time PCRs. (D) Expression levels of DNMT3A in E4 cells transduced with lentiviruses harboring *Dnmt3a*-3xFlag and ChIP assays performed on cells treated with indicated conditions using Flag MS2 magnetic beads to pull down DNMT3A-3xFlag proteins. Cells were transduced and selected by puromycin (2 µg/ml) for 3 days. Before ChIP, cells were treated with GSK-J4 (5 µM) for 48 hours. Error bars: SEM of 3 separate quantitative real-time PCRs. (E) DNA demethylases TET1 and TET2 are involved in HIV-1 reactivation in Jurkat T cells. Reactivation of latent HIV-1 induced overnight by SAHA (1 µM) or TNF-α (10 ng/ml) in TET1 or TET2 knocked-out E4 cells. Error bars: SEM of 7 separate replicates. One-way ANOVA, $p < 0.05$ Bonferroni posttests, n = 7 * p $< 0.05$ *** p$<0.001$.

In a complementary approach, we directly measured the levels of 5'-methylcytosine at the HIV-1 provirus in a E4 clone expressing UTX-1 shRNA by MeDIP coupled to qPCR. The primers used for qPCR (Fig 5A) were designed to specifically amplify the 5' CpG, NCR CpG and Env CpG islands [67]. However, the Nuc-2 primers used in this study only covered the downstream region of NCR CpG cluster. Compared to HIV-1 from control cells, HIV-1 from the shUTX knock down cells contained much higher levels of 5'-methylcytosine. Similarly, HIV-1 from E4 cells treated with GSK-J4 also contained higher levels of DNA methylation at the 5' CpG and Nuc-2 clusters than HIV-1 proviruses from untreated cells (Fig 5C).

Taken together, our data demonstrate that either the depletion of UTX by shRNA or inhibition of its demethylase activity by GSK-J4 greatly increased the DNA methylation of HIV-1 proviruses.

## DNMT3A is recruited to the HIV-1 LTR after inhibition of UTX by GSK-J4

To further study the interplay between UTX and DNA methylation, we ectopically expressed the transgene DNMT3A-3xFlag. In this construct, 3xFlag was introduced at the C-terminal of DNMT3A which allowed us to perform ChIP assays using the Flag M2 antibody to measure the enrichment of DNMT3A at HIV-1 5'LTR. We chose this approach because no suitable antibodies for ChIP against the endogenous DNMT3A were available.

Western blots performed on E4 cells transduced with *Dnmt3a*-3xFlag lentiviruses showed greatly increased levels of DNMT3A (Fig 5D). ChIP assays showed that treatment of these cells with 5 µM GSK-J4 for 48 hours resulted in significant recruitment of DNMT3A to HIV-1 5' CpG and Nuc-2 clusters (Fig 5D), but not to the promoters of two control genes, TNFA and GAPDH. Finally, as a complementary approach, we also knocked out the DNA demethylases TET1 or TET2 using CRISPR-Cas9. Knock out of either gene partially inhibited HIV-1 reactivation in Jurkat cells after reactivation with SAHA or TNF-α (Fig 5E). In conclusion, inhibition of UTX results in specific recruitment of DNMT3A to HIV-1, which in turn elevates DNA methylation of the HIV-1 provirus and enhances HIV-1 silencing.

## Inhibition of UTX accelerates silencing of HIV-1 in primary cell models of HIV-1 latency

Epigenetic silencing in primary cells involves additional important mechanisms that are not well represented in the transformed Jurkat cell background. Specifically, we showed previously that both the EZH2 and EHMT2 histone methyltransferases strongly restrict HIV-1 transcription in primary T-cell models for HIV-1 latency and in resting memory T-cells isolated from HIV-1 infected patients [22]. However, EZH2 is the dominant histone methyltransferase in Jurkat T-cells and EHMT2 makes only a small contribution to silencing in Jurkat T-cells [22]. We therefore extended our study of the role of the H3K27 demethylase activity of UTX on

HIV-1 silencing and reactivation using the well-established QUECEL (Th17 cells) primary cell model of HIV latency [21, 68] (Fig 6).

As outlined in Fig 6A, naïve CD4+ T cells obtained from 4 different donors were polarized to a Th17 phenotype and then infected with a single round HIV-1 reporter expressing d2EGFP, following the QUECEL protocol [21]. Three days after HIV-1 infection (Day 0), the cells were grown in the presence of 1 μM of GSK-J4 for 4 days, with fresh media containing the drug added after 2 days. The addition of GSK-J4 increased global levels of H3K27me3 more than two-fold on average in 4 different donors (Fig 6B).

At Day 0, the infected cells were transferred to media containing low concentrations of IL-2 (15 IU/ml), and IL-23 (12.5 μg/ml). The levels of d2EGFP expression were measured by flow cytometry at days 4, 6, or 8 to monitor HIV silencing. In the presence of these low concentrations of cytokines, cells gradually enter quiescence leading to silencing of HIV-1 transcription (Fig 6C). The extent of HIV silencing is significantly enhanced by GSK-J4. At Day 4, the average d2EGFP expression from 4 donors was 46.5% relative to day 0 (ranging from 35.5% to 54.6%) while the average d2EGFP expression from the untreated cells was 59% (ranging from 41.6% to 68.3%). The differences in the average d2EGFP expression between cells treated with GSK-J4 and untreated cells progressively increased by Day 6 (34.5% vs 54.2%) and Day 8 (26.2% vs 47.9%) (Fig 6C). The time for HIV silencing is shown as averaged data in Fig 6C, and the kinetics of each individual donor is presented in S7 Fig. To confirm that the populations of GFP-negative cells were latently infected, the cell populations at Day 8 were reactivated using a combination of SAHA (1 μM) and IL15 (10 ng/ml), which is a potent reactivation condition for HIV in primary T-cells [21] (Fig 6C).

In parallel, HIV-1 DNA methylation was measured by MeDIP at Day 4. Prior to treatment with GSK-J4, 5'-methycytosine was not detected at the 5' CpG and NCR CpG clusters in each of the 4 different donors, suggesting that there was no DNA methylation of HIV-1 at these regions under these conditions. Substantial increases in the levels of DNA methylation at these CpG clusters were detected after treatment with 1 μM GSK-J4 (Fig 6D). The increases in DNA methylation of HIV-1 was statistically more significant at the 5' CpG region, the region upstream of HIV-1 TSS which also overlaps with a wide variety of transcription factor binding sites, than in the other regions.

Since UTX is necessary for the reactivation of latent HIV-1 in Jurkat T cells, its role in the reactivation of latent HIV-1 in primary Th17 cells was also investigated by pretreatment of the cells with increasing concentrations of GSK-J4 for 24 hours prior to reactivation. The drug has no toxicity under these conditions as shown by the $IC_{50}$ and $CC_{50}$ values of GSK-J4 on Th17 primary T cells treated for 48 hours (S4 Fig). Treatment of the control cells with Dynabeads Human T-Activator CD3/CD28 to activate the TCR, reactivated more than 50% of latent viruses (from 21% to 77% d2EGFP). However, only 17% and 4% of latent viruses were reactivated in cells treated with 2 μM and 4 μM of GSK-J4, respectively (S8 Fig). Similar results were also obtained when cells were treated with a combination of SAHA and IL-15. Quantification of data from 3 different experiments on the same donor also showed that GSK-J4 significantly inhibited the reactivation of latent HIV-1 in Th17 primary cells (Fig 6E). In conclusion, UTX inhibits the entry of HIV-1 into latency and is required for the reactivation of latent HIV-1 in primary Th17 cells.

## GSK-J4 suppresses HIV-1 reactivation in CD4+ memory T cells isolated from HIV-1 infected donors

To investigate the impact of GSK-J4 on reactivation of latent HIV-1 in CD4+ memory T cells obtained from patients, we obtained blood samples from 5 well-suppressed HIV-1 donors and

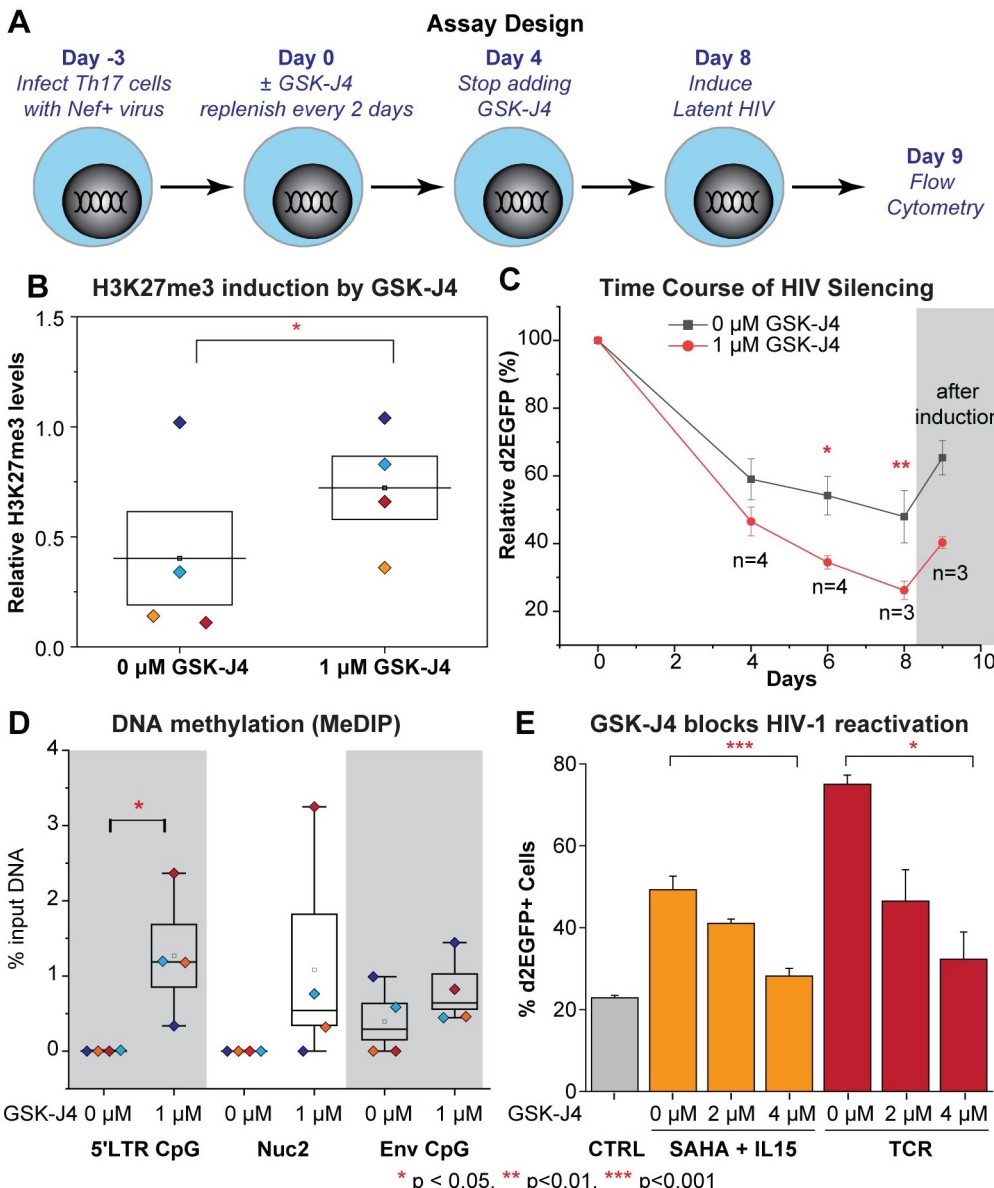

**Fig 6. Inhibition of UTX by GSK-J4 promotes the silencing of HIV-1 in Th17 primary cells and elevates HIV-1 DNA methylation.** (A) Schematic of experimental design. (B) Relative levels of H3K27me3 compared to histone H3 levels quantified from Western blot on treated Th17 cells from 4 donors at Day 4. One-tailed, paired t-test, * p<0.05. (C) Silencing kinetics of HIV-1 in Th17 cells from 4 different donors in the presence of 0 μM (vehicle) or 1 μM of GSK-J4. Error bars: SEM of at least 3 separate biological replicates. Two-way ANOVA, Bonferroni posttests, * p<0.05 ** p<0.01. (D) MeDIP measuring the levels of methylated cytosine of HIV-1 at Day 4. Error bars: SEM of 4 different donors. One-tailed, paired t-test, n = 4, * p<0.05. (E) GSK-J4 inhibits the reactivation of latent HIV-1 by SAHA & IL15 or TCR stimulation in Th17 primary cells. Cells were pretreated for 24 hours with GSK-J4, then further stimulated overnight with SAHA & IL15 or Human T-Activator CD3/CD28 Dynabeads with the ratio of 25 μl of beads per 1 million cells. Quantification of HIV-1 reactivation under the indicated conditions. Note that only viable cells were chosen for HIV-1 reactivation measurement. Error bars: SEM of more than 3 separate replicates. One-way ANOVA, Bonferroni posttests n >3.

frozen PBMCs from 1 donor who underwent leukapheresis. The purified memory cells were treated with 5 μM GSK-J4 for 3 days (Fig 7A). The cells were then treated with Dynabeads Human T-Activator CD3/CD28 over night to reactivate latent HIV-1. We measured HIV-1 reactivation by the EDITS assay which is a nested PCR assay measuring the production of spliced env mRNA, an RNA species only found in fully activated T-cells [68] (Fig 7B). GSK-J4 significantly inhibited the reactivation of latent HIV-1 in CD4+ memory T cells, with only about 30% of HIV env mRNA+ cells observed compared to untreated cells. As a control, the expression level of CD69 was measured by FACS to provide a marker for T cell activation in the entire population of latently infected and bystander cells. As expected, greater than 80% of the activated cells became CD69+ (Fig 7C). This data also demonstrated that the inhibition of latency reversal by GSK-J4 was specific, since bystander cells treated with GSK-J4 expressed comparable levels of CD69 compared to the untreated cells after TCR stimulation (Fig 7C). There was also a small, but statistically significant, increase in H3K27me3 expression in the treated cells, and a correlation between variation in the inhibition levels of GSK-J4 on HIV-1 reactivation and H3K27me3 levels between the different donors (Fig 7D). Differences in the reactivation levels of untreated and GSK-J4 treated latent viruses were not due to toxicity of the compound because there was no significant decrease in cell viability (Fig 7E). Therefore, in confirmation of the results obtained using the primary T cell model, reactivation of latent HIV-1 in CD4+ memory T cells from HIV-1 infected donors was also blocked by GSK-J4.

## GSK-J4 induces transient HIV-1 DNA hypermethylation in CD4+ memory T cells isolated from HIV-1 infected donors

The ability of GSK-J4 to induce DNA methylation of latent HIV-1 was investigated in CD4 + memory T cells. Because of the low frequency of latently infected cells in the patient samples, HIV-1 DNA methylation was measured by targeted next-generation bisulfite sequencing on proviral DNA isolated from the same donors used in the latency reversal experiments in Fig 7.

The 5'LTR CpG (or 5' CpG) contains nine different CpG dinucleotides. Their positions relative to HIV-1 transcription start site are shown in Fig 8A. Due to limitations in the amount of HIV-1 DNA, we only investigated DNA methylation of the 5'LTR CpG cluster, which is the most relevant to transcriptional control of HIV-1 [69]. As shown in Fig 8B, GSK-J4 increased average HIV-1 DNA methylation (CpG 1 to 9) for all tested donors and CpG1-specific methylation for 5 out of 6 donors. Untreated donors showed barely detectable levels of DNA methylation of latent HIV-1 (mean of the average levels was 0.7%), consistent with the paucity of HIV-1 DNA methylation at the same region reported from many studies [69–72]. Upon GSK-J4 treatment, a 1.3 to 10-fold (average of 3-fold) increase in the average DNA methylation was observed from all tested donors. This is inversely correlated with inhibitions of HIV-1 reactivation measured by EDITS assays in the same donors (Figs 7B and 8B). Interestingly, out of the nine CpG islands analyzed, we consistently detected the highest increase of methylation at CpG1 in 5 out of 6 donors (Fig 8C), while changes of methylation at the other CpG islands were very modest (S9 Fig). An average of 4-fold increase in the DNA methylation levels was observed at CpG1.

We next investigated whether DNA methylation at 5'LTR of HIV-1 induced by GSK-J4 could be maintained after GSK-J4 withdrawal. CD4+ memory T cells from a different batch of three HIV-1 infected donors were treated with 5 μM of GSK-J4. After 72 hours, a portion of cells was collected for next-generation bisulfite sequencing. Media containing GSK-J4 from the remaining cells was washed away and cells were cultured for an additional 72 hours in media without GSK-J4. The DNA methylation levels at 5'LTR of HIV-1 from four different donors before and after GSK-J4 removal were presented in Fig 8C. Similar to what we showed

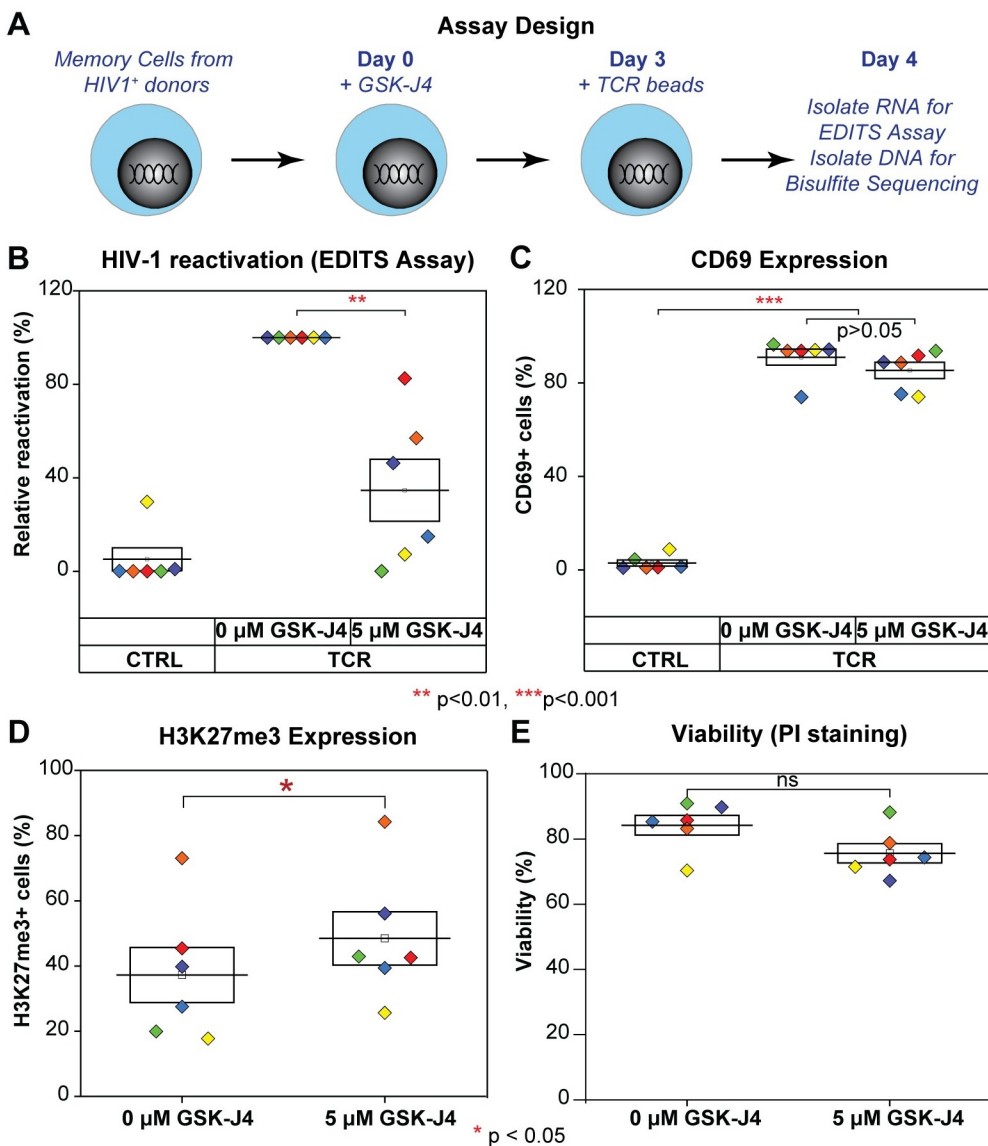

**Fig 7. GSK-J4 inhibits reactivation of latent HIV-1 in CD4 memory T cells isolated from well suppressed HIV-1 infected donors.** (A) Schematic diagrams describing the experimental designs. (B) Inhibition of latent HIV-1 reactivation in CD4+ memory T cells from HIV-1 infected patients on cARTs by UTX inhibitor. CD4+ memory T cells isolated from HIV-1 infected patients were treated for 3 days with indicated concentrations of GSK-J4. Cells were left untreated or further treated with human T-Activator CD3/CD28 Dynabeads overnight. Levels of spliced HIV-1 env mRNA were measured by EDITS assays. Levels of relative reactivation were normalized to the levels of spliced HIV-1 env mRNA induced by human T-Activator CD3/CD28 Dynabeads (presented as 100%) for each donor. (C) Expression levels of CD69 from treated CD4+ memory T cells measured by FACS using CD69 antibody. One-way ANOVA $p < 0.005$, Bonferroni posttests * $p<0.05$ ** $p<0.01$ *** $p<0.001$ were performed for both Fig 7B & 7C. (D) Intracellular H3K27me3 levels of treated CD4+ memory T cells measured by FACS using a H3K27me3 antibody. Cells were stained with Alexa Fluor 647-conjugated trimethyl histone H3 (Lys27) antibody (12158, Cell Signaling) and analyzed by FACS as described previously [22]. (E) Viability of cells by PI staining. After drug treatment, cells were stained with PI at the final concentration of 5 μg/ml for 5 minutes, then analyzed by FACS. Two-tailed paired t-test was performed for both Fig 7D & 7E, * $p<0.05$, ns: not statistically significant.

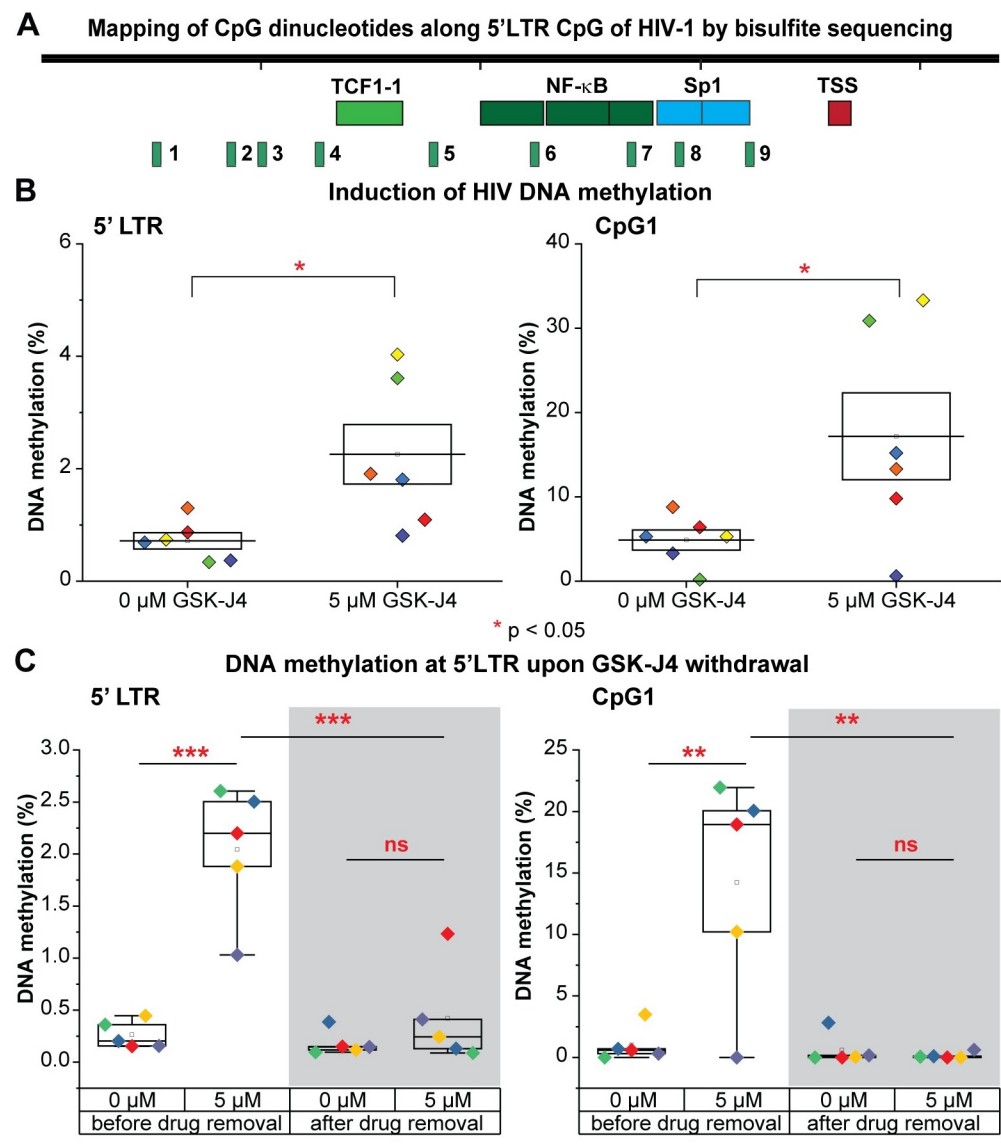

**Fig 8. Temporary induction of DNA methylation at 5'LTR of latent HIV-1 in CD4 memory T cells upon the inhibition of UTX by GSK-J4.** (A) The map of CpG dinucleotides (numbered from 1 to 9) relative to HIV-1 transcription start site along the 5'LTR CpG cluster. (B) The average DNA methylation (as %) of CpGs and the percentage DNA methylation at CpG island 1 of the 5'LTR CpG cluster from CD4+ memory T cells treated with the indicated concentration of GSK-J4. (C) The average DNA methylation of CpGs and the percentage DNA methylation at the CpG island 1 at the 5'LTR CpG cluster of HIV-1 from CD4+ memory T cells induced with GSK-J4 before or after GSK-J4 removal. Experiments were performed in single replicates with three different donors and in duplicates with one additional donor; One-tailed paired t-tests were performed on all experiments, * p<0.05, ** p<0.01, *** p<0.001, ns: not statistically significant. Note: different HIV-1 donors were utilized for this experiment than those utilized in Fig 8B.

previously, DNA methylation at 5'LTR of HIV-1 was significantly increased with an average of 4 fold by GSK-J4 treatment. After GSK-J4 removal, DNA methylation levels at HIV-1 5'LTR were restored back to the initial very low levels seen in untreated cells. As noted previously the most substantial changes in DNA methylation at the HIV LTR were at CpG1, and this returned to control values after wash out of the drug (S10 Fig).

A replicate experiment performed in duplicate using a single donor, which measures the impact of GSK-J4 on HIV reactivation with EDITs assay, DNA methylation, CD69 expression, H3K27me3 expression, and cell viability and the rapid reversion of cells to an untreated phenotype after wash out of the drug is shown in S11 Fig. Before GSK-J4 withdrawal, levels of HIV-1 reactivation measured by EDITs assay were completely inhibited with GSK-J4 treatment. After GSK-J4 withdrawal, levels of HIV-1 reactivation in cells treated previously with GSK-J4 were restored up to 80% compared to those from cells untreated (S11A Fig). Levels of CD69 did not vary much between untreated cells and cells treated with GSK-J4 before or after GSK-J4 withdrawal (S11B Fig). Somewhat surprisingly, we see a somewhat persistent increase in the global levels of H3K27me3 in cells treated with GSK-J4 after GSK-J4 withdrawal, indicating that turnover rate of H3K27me3 was more than 72 hours (S11C Fig). The viability of cells treated with GSK-J4 was slightly decreased after drug withdrawal (S11D Fig). Levels of HIV-1 DNA methylation at 5'LTR or CpG1 were highly increased in cells treated with GSK-J4 before drug withdrawal; however, after drug withdrawal, DNA methylation of HIV-1 in cells treated with GSK-4 was restored to levels similar to untreated cells at both at the 5'LTR and CpG1 (Fig 8B and 8C and S11E and S11F Fig). This indicated an inverse correlation of HIV-1 reactivation and DNA methylation at 5'LTR of HIV-1. Taken together, the data confirmed the transient effect of GSK-J4 on the inhibition of HIV-1 reactivation and induction of HIV-1 DNA methylation; even though its effect on inhibition of UTX activity might still be maintained globally 72 hours after drug removal. However, H3K27me3 at HIV-1 has been known not to resist to reactivation by strong stimulus like TCR. In addition, this is a global measurement of H3K27me3 in cells but not a measurement of H3K27me3 at HIV-1 chromatin.

## Discussion

### UTX is required for HIV-1 reactivation

UTX is a specific H3K27 demethylase, associated with the MLL2 and MLL3 containing complexes [73], which are also responsible for H3K4 methylation. The combined reduction of H3K27me3 levels and increase in H3K4me3 levels provides a key step in the transcription activation of bivalent promoters. Our earlier studies clearly demonstrated that EZH2, the enzymatic subunit of the PCR2 complex, required for H3K27me3 formation, is an essential silencing factor for HIV-1 transcription [22]. We reasoned that blocking UTX might enhance H3K27me3 accumulation at the HIV-1 provirus and block proviral reactivation. Consistent with this hypothesis, Zhang et. al [48] showed that UTX was an activator for the transcription of HIV-1 LTR-driven luciferase in TZM-bl cell, a Hela cell line harboring an integrated copy of LTR-luciferase [74].

Using a wide variety of approaches to block UTX function in multiple HIV-1 latently infected Jurkat T-cell lines, primary cells and latently infected cells obtained from patients, we demonstrate here that UTX is essential for reactivation of latent HIV-1. Depletion of UTX by shRNA (Fig 1) or inhibition of its activity by GSK-J4 prevents the transcription of latent provirus both in Jurkat T cells (Fig 3A) and resting CD4+ T cells cultured *ex vivo* (Fig 6E).

Importantly, pharmacological inhibition of UTX enzymatic activity by GSK-J4 performed on CD4+ memory T cells isolated from well-suppressed HIV-1 infected donors significantly reduces the induction of HIV-1 reactivation by TCR stimulation (Fig 7B). GSK-J4 also accelerates the silencing of provirus in primary Th17 cells (Fig 6A–6C).

Finally, the inhibition of HIV-1 reactivation by GSK-J4 or UTX depletion by shRNA is observed in cells infected with wild type, H13L Tat, or Nef-deleted mutants of HIV-1 (Fig 1). H13L is a fully functional Tat, that readily supports HIV-1 reactivation in different stimuli (Fig

1B). This suggests that UTX activation of HIV-1 transcription is due to epigenetic transcriptional activation of the HIV LTR and is independent of Nef or Tat function.

## UTX is required to resolve the bivalent chromatin at the 5'LTR of latent HIV-1 proviruses

Genome-wide chromatin immunoprecipitation studies have described bivalent genes, characterized by the co-occurrence of H3K27me3, a repressive mark, and H3K4me3, an activating mark, within a single nucleosome [75, 76]. Bivalent genes are therefore maintained in a poised or reversibly silent state that could be promptly and forcefully activated with an appropriate signal [49]. However, the stimulus signal threshold required for full activation of bivalent genes is thought to be higher than other genes [49].

The 5' LTR of latent HIV-1 in T cells closely resembles cellular bivalent genes [49] and is characterized by the presence of H3K4me3 and H3K27me3 (Fig 3B), restricted transcription by PRC2 [22], and high enrichment of CpG dinucleotides [77] but low methylation of DNA regions surrounding the transcription start site (Figs 5C and 8B) [78, 79]. Bivalency is also consistent with robust reactivation of HIV and the extensive limitation of viral transcription elongation (Figs 2 and 3) [20, 22, 23, 47, 68].

UTX recruitment to the HIV-1 LTR is enhanced when latent HIV-1 is reactivated by TNF-α (Fig 2), and alters the balance between H3K27me3 and H3K4me3 at the LTR after reactivation (Fig 3B). Inhibition of UTX by GSK-J4 blocks the H3K27me3 reduction and H3K4me3 increase at the HIV-1 LTR (Fig 3B), and correlates with the inhibition of HIV-1 reactivation in Jurkat T cell (Fig 3A).

This entire process is orchestrated by UTX, which combines its intrinsic H3K27 demethylase activity and its association with the MLL2/MLL3 complexes. The requirement for UTX to induce transcription of cellular bivalent genes has been well documented [35, 36], and is similar to HIV-1, as reported in this study. We confirmed the functional significance of H3K27me3 and H3K4me3 for HIV transcription using ectopic expression of H3.3 variants carrying mutations at K27 and K4. These experiments showed that HIV-1 reactivation mediated by H3.3 K27M is inhibited by H3.3 K4M, consistent with the bivalency of its 5'LTR (Fig 4). Therefore, our ectopic expression experiments suggest that both H3K27me3 displacement and H3K4me3 deposition on the same nucleosome, and possibly on the same histone H3 subunits, are required for efficient reactivation of latent HIV-1. The bivalency of the HIV promoter may also explain why permanent suppression of the provirus is difficult to achieve.

## Transient DNA methylation of latent HIV-1 is induced by inhibition of UTX

The interplay between H3K27me3 and DNA methylation has not been fully elucidated. Genome-wide studies mapping overlapping sites of H3K27me3 by ChIP-seq and methylated CpG microarray data reveal a mutually exclusive relationship between these two marks at CpG promoters [80–82]. However, some studies report the correlation of H3K27me3 and DNA methylation at the promoters of a subset of genes [78, 80]. More recently, by bisulfite sequencing of chromatin immunoprecipitated DNA, Statham et al. has reported that H3K27me3 can co-occur with either unmethylated or methylated DNA [83]. In addition, DNA hypomethylation resulted from induced H3K27me3 reduction is observed at promoters of many genes in K27M mutant pediatric high-grade gliomas [60]. In addition, PRC2 recruitment and the presence of H3K4me3 have been shown to provide cellular promoters protection from DNMT-mediated DNA methylation [58]. These findings indicate that change in the levels of H3K4me3 and H3K27me3 at specific promoters affects promoter DNA methylation.

Using biochemical and MeDIP assays, we demonstrate that depletion of UTX expression by shRNAs or pharmacological inhibition of its H3K27 demethylase by GSK-J4 results in increased DNA methylation of latent HIV-1 in Jurkat T cells. DNA methylation occurs at the 5'CpG and NCR CpG clusters, surrounding the HIV-1 TSS (Fig 5). Similar DNA methylation patterns were seen in Th17 primary cells after treatment with GSK-J4 (Fig 6D). In addition, targeted next-generation bisulfite sequencing assays performed on CD4+ memory T cells isolated from well-suppressed HIV-1 infected donors also demonstrate a significant induction of DNA methylation at the 5'LTR CpG cluster when proviruses are treated with GSK-J4 (Fig 8B and 8C). The extent of DNA methylation at the HIV 5'LTR under these conditions in our study was comparable to those acquired from specific recruitment of a chimeric zinc finger DNMT1 to HIV-1 5'LTR performed in Jurkat T cells [84].

Previous reports have indicated a link between H3K27me3 and DNA methylation pathways [78, 80]. Bender et al. described that reduced H3K27me3 in K27M mutant pediatric high-grade gliomas results in DNA hypomethylation in many genes [60]. An earlier study demonstrated the correlation between DNA hypermethylation and loss of bivalent chromatin [85]. Though the precise molecular mechanisms involved have yet to be clarified, it is likely that H3K27me3 retention and H3K4me3 loss at HIV-1 5'LTR caused by UTX depletion or GSK-J4 (Figs 2B and 3B) mediates the recruitment of DNA methyltransferases, including DNMT3A, to HIV-1 (Fig 5C and 5D). This hypothesis is supported by several studies showing that elevation of H3K27me3 or retention of H3K27me3 and loss of H3K4me3 is correlated with DNA hypermethylation [60, 83, 85]. Recruitment of de novo DNA methyltransferase DNMT3A to the 5'LTR CpG upon inhibition of UTX/JMJD3 by GSK-J4 likely results in induction of DNA methylation at that region; however, the involvement of DNMT1 in this process could not be refuted.

We consistently observed that the largest increase of methylation at the HIV 5'LTR occurs at CpG1, a region that does not overlap with the majority of transcription factor binding sites. We hypothesize that methylation at this CpG1 indirectly prevents the recruitment of transcription activators to the nearby locations by mediating recruitment of DNMTs. However, further experiments will be required to reveal how methylation at this CpG dinucleotide negates HIV-1 transcription.

## Implications for the "block and lock strategy" for a functional HIV cure

The inhibitory effect of GSK-J4 on latent HIV-1 reactivation in Jurkat T cells (Fig 3A), Th17 primary T cells (Fig 6E), and CD4+ memory T cells from HIV-1 infected donors (Fig 7B), demonstrates its potential as an agent for blocking HIV-1 rebound in infected patients. In addition to its inhibition of UTX, the ability of GSK-J4 to induce 5'LTR DNA methylation of latent HIV-1, although only temporary, is intriguing as it relates to "block and lock" studies of the irreversible silencing of latent HIV-1.

Among gene silencing mechanisms, DNA methylation at promoters has been suggested as a mechanism for the stable suppression of gene expression. DNA methylation at promoters is involved in imprinted X inactivation [86], and in suppression of human endogenous retroviruses (HERVs) [87]. Unlike the heavy DNA methylation at the LTR of HERVs (more than 30% [88]), DNA methylation at 5'LTR CpG cluster of latent HIV-1 is scarcely detected in this study (the average observed DNA methylation is less than 1%—Fig 8B) and other studies [69–72], indicating that latent HIV-1 is actually maintained in a poised state rather than a stably repressed state like imprinted genes on inactive X chromosome or HERVs. It is unclear what determines the differences of DNA methylation observed between latent HIV-1 and HERVs. Possession of a defective genome cannot account for this variance given that only 45% of latent HIV-1 clones from patients have large deletions [72].

We hypothesize that DNA methylation at the HIV LTR is highly dynamic. Two features of HIV-1 likely protect HIV-1 from long-term DNA methylation: first, the bivalency of its LTR and, second, the active involvement of TET proteins in DNA demethylation of sites in the HIV-1 LTR (Fig 5E). PRC2 recruitment and H3K4me3 presence have been shown to provide gene promoters protection from DNMT-mediated DNA methylation [58]. Reduction of H3K4me3 levels at HIV-1 LTR by UTX knock down or GSK-J4 treatment erases this protection from DNMT and permits the recruitment of DNMT3A to HIV-1, which in turn results in elevated DNA methylation of HIV-1. In addition, the active DNA demethylation of HIV-1 by TETs may be another factor protecting latent HIV-1 LTR from DNA hypermethylation. Given the essential roles of TET proteins in the development, proliferation, and differentiation of T cells, including CD4+ T cells [89–91], it is not surprising that either TET1 or TET2 knock out partially inhibited HIV-1 reactivation in Jurkat cells (Fig 5E). We hypothesize that the involvement of multiple TET proteins leads to rapid HIV-1 demethylation. Unfortunately, because of toxicity effects we were unable to simultaneously knockout TET1 and TET2 to test this model.

Our study is the first one to report that DNA methylation of latent HIV-1 can be induced by a small molecule inhibitor like GSK-J4. It is a measure of the great potency of these epigenetic restrictions that under these conditions, latency reversal due to TCR activation was blocked by over 70% in patient cells, with some donors showing nearly complete blocks to reactivation. Nonetheless, these powerful epigenetic blocks did not lead to long-term epigenetic silencing. It is notable that GSK-J4 only temporarily promotes DNA methylation at 5'LTR of HIV-1 by recruitment of DNMT3A. A rapid turnover of DNA methylation at the HIV LTR could therefore explain the extremely low levels of DNA methylation measured at 5' LTR of latent HIV-1 in comparison to those detected at imprinted X genes or HERVs.

We are therefore skeptical that the long-term HIV silencing following treatment of cells by didehydro-cortistatin A is primarily due to epigenetic silencing [14, 17–19]. Alternative mechanisms that could explain the very intriguing data that dCA reduces functional HIV reservoirs could include the selective loss of cells carrying proviruses during dCA exposure or reactivation, or long-term inhibition of a cellular cofactor required for transcription, such as the mediator complex. These and other possibilities need to be evaluated as potential explanations for these provocative observations.

In summary, our study documents the essential role of UTX as an activator for HIV-1 transcription as well as a mediator for maintaining the bivalent chromatin at the 5' LTR of latent HIV-1 proviruses. We have identified GSK-J4, a H3K27 demethylase inhibitor, as a potential agent for blocking latent HIV-1 reactivation in T cells. Unexpectedly, GSK-J4 has been shown to be able to induce DNA methylation of latent HIV-1, but unfortunately, the induction of DNA methylation by GSK-J4 on HIV-1 was only maintained in the presence of GSK-J4 and could not persist after GSK-J4 removal. Methods to further enhance DNA methylation at the HIV LTR should be further explored as part of a "block and lock" strategy. These could include targeted recruitment of DNMTs to HIV-1 [84, 92, 93], and induction of DNA methylation by piRNA production [94]. However, since DNA methylation by DNMT3A and DNA demethylation by TET1 and TET2 are highly dynamic, there remain formidable intrinsic barriers to achieving the permanent epigenetic silencing of HIV proviruses.

## Materials and methods

### Ethics statements

Blood samples used in this work were obtained under University Hospitals Clinical Research Center Institutional Review Board (IRB) consent 12-11-33. Formal written consent was

obtained for all blood samples obtained from the CWRU/UH Clinical Core under this protocol.

## Cell lines and cell culture reagents

E4, 2D10, and 3C9 latently HIV-1 infected Jurkat T cell lines were used [23]. E4 and 2D10 cells were infected with Nef deleted HIV-1, while 3C9 cell line contained Nef + HIV-1. Cells were cultured in HyClone RPMI medium with L-glutamine, 5% fetal bovine serum (FBS), penicillin (100 IU/ml), streptomycin (100 μg/ml) in 5% $CO_2$ at 37˚C. Primary T cells were cultured in RPMI medium supplemented with 10% FBS, 1 mg/ml normocin (Invivogen, ant-nr), and 25 mM HEPES pH 7.2.

## VSV-G pseudotyped HIV-1 virus production

VSV-G pseudotyped HIV-1 was produced as previously described using the Thy1.2-d2EGFP--Nef-pHR' vector (which expresses Thy1.2, Nef, and d2EGFP), as well as the pdR8.91 and VSV-G vectors [23]. Cells were infected with HIV-1 by spinoculating with viruses at 3480 rpm for 1.5 hrs at room temperature.

## UTX shRNA, TET1/TET2 CRISPR-Cas9, *H3.3 mutant*, and *Dnmt3a*-3xFlag constructs

Two sequences of UTX shRNAs were used: UTX-1(TGGAACAGCTCCGCGCAAATA) and UTX-2(TGCACTTGCAGCACGAATTAA). shRNAs were cloned into pLVTHM plasmid which was a gift from Didier Trono (Addgene plasmid # 12247). We replaced GFP from pLVTHM with mCherry for selection. HA tagged *H3.3* wild type (wt) and K27M lentiviral vectors were received from the previous study [57]. *H3.3* K4M and K4M-K27M mutants were made by site-directed mutagenesis from the *H3.3* wt and K27M constructs. *Dnmt3a*-3xFlag lentiviral construct was ordered from VectorBuilder (vector ID: VB180918-1067xya). Guide RNAs targeting TET1 (GCATGGAAGAGTCCTCTCTC and AAAGGCCTGTCCTAG GAAAG) or TET2 (TTCTGGGTGTAAGCTTGCCT and GGTTGATACTGAAGAATTGA) were cloned into the lentiCRISPR v2 (Addgene #52961). Lentiviral production and transduction of cells with viruses were described previously [22].

## Western blot

The following antibodies were used for Western blot: UTX (A302-374A, Bethyl), β-actin (sc-47778, Santa-Cruz), Histone H3 (ab1791, Abcam), Histone H3K27me3 (ab6002, Abcam), Histone H3K4me3 (ab8580, Abcam), Histone H3K4me2 (ab1220, Abcam), HA tag (sc805, Santa-Cruz), Flag M2 (F1804, Sigma), and α-tubulin (ab4074, Abcam). Western blot was performed as described previously [23]. Fifty μg of total cell lysate per lane was loaded.

## ChIP-qPCR analysis

Chromatin immunoprecipitation (ChIP) was performed as previously described using the Pierce Agarose ChIP Kit (Thermo Scientific) [22, 23]. An IgG control antibody is included in the kit which we routinely included in our ChIP assays. The signals generated by IgG antibody were subtracted from the signals obtained with specific antibodies. The following antibodies were used for ChIP: RNAP II (17–672, Millipore) or (sc-899, Santa Cruz), EZH2 (17–662, Millipore), Histone H3 (ab1791, Abcam), Histone H3K4me3 (ab8580, Abcam), Histone H3K27me3 (ab6002, Abcam), and UTX antibody (A302-374A, Bethyl). The Flag M2 magnetic beads (M8823, Sigma) were used to immunoprecipitate DNMT3A-3xFlag in (Fig 5) following

the same protocol for ChIP. The percentage of input method was used to calculate the enrichment of proteins at specific regions of HIV-1 genome. Primers were used for HIV-1 DNA amplification: HIV-1 Nuc-0: -390F, ACA CAC AAG GCT ACT TCC CTG A, and -283 R, TCT ACC TTA TCT GGC TCA ACT GGT; HIV-1 promoter: -116 F, AGC TTG CTA CAA GGG ACT GGT, and +4 R, ACC CAG TAC AGG CAA AAA GCA G; HIV-1 Nuc-1: +30 F, CTG GGA GCT CTC TGG CTA ACT A, and Nuc-1- R, TTT CAA GTC CCT GTT CGG GCG and HIV-1 Nuc-2: +283 F, GAC TGG TGA GTA CGC CAA AAA T, and +390 R, TTT CCC ATC GCG ATC TAA TTC. Primers were used for amplification of GAPDH and TNFα promoters: TNF-F: CCCCCTCGGAATCGGA, TNF-R: GAGCTCATCTGGAGGAAGCG, GAPDH-F: CGGTGCGTGCCCAGTT, GAPDH-R: CCCTACTTTCTCCCCGCTTT.

## Methylated DNA immunoprecipitation (MeDIP)

DNA was isolated by Qiagen DNeasy kit, then sonicated for 20 minutes (30 second ON/30 second OFF) to generate fragments with the size of 500 bp. Two μg of DNA was heat-denatured for 10 minutes at 95˚C to produce single-stranded DNA. Two μg of antibody against 5-methyl cytosine (ab10805, Abcam) was used to immunoprecipitate methylated DNA fragment for overnight. Bound DNA was washed thoroughly and eluted for quantitative measurement by qPCR. The following primers were used for qPCR: 5'LTRCpG-F: GAAGTGTTAGAGTG GAGGTTTGA, 5'LTRCpG-R: CAGCGGAAAGTCCCTTGTAG, EnvCpG-F: TTTGTTCCTTGGGTTCTTGG, EnvCpG-R: TGGTGCAAATGAGTTTTCCA.

## Production of HIV-1 latently infected primary cells and virus reactivation

HIV-1 latently infected Th17 primary cells (QUECEL) were produced following a previously described procedure [68] and using Thy1.2-d2EGFP-Nef-pHR' construct. Reactivation of proviruses was performed by incubating cells overnight with DynabeadsHuman T-Activator CD3/CD28 (25 μl/10^6 cells) or a combination of SAHA (1 μM) and IL15 (10 ng/ml).

## Treatment of E4 cell line with 5'-Azacytidine (5-AZC)

5-AZC was purchased from Sigma (A2385) and dissolved in 50% acetic acid. Cells were pretreated with 1 μM 5-AZC for 72 hours then left untreated or treated further with a combination of anti-CD3 (125 ng/ml) and anti-CD28 (500 ng/ml), 500 nM of SAHA, or 1 ng/ml of TNFα overnight. HIV-1 reactivation in the cells was measured by FACS. Fold of HIV-1 reactivation was calculated by normalizing the levels of d2EGFP expression after drugs treatment to those obtained from the 5-AZC untreated conditions.

## Treatment of HIV-1 latently infected Jurkat cells and primary cells with inhibitors

GSK-J4 was purchased from Selleckchem (S7070) and dissolved in DMSO. Cells were treated with increasing concentrations of GSK-J4. Cell viability was measured by propidium iodide (5 μg/ml, 10008351-Cayman) or calcein AM viability dye (65-0854-39-eBioscience) staining.

## RNA induction assay (EDITS assay)

CD4+ memory T cells isolated from HIV-1 infected donors were treated with GSK-J4 for 3 days. Then cells were stimulated overnight with Dynabeads Human T-Activator CD3/CD28, followed by staining with anti-CD69 (1:100 dilution, 557745 –BD Pharmingen) and FACS analysis. EDITS assays to measure the reactivation of latent HIV-1 in CD4+ memory T cells

isolated from well suppressed HIV-1 infected donors were performed as described previously [68].

## Targeted next-generation bisulfite sequencing

PBMCs were purified from fresh blood samples, which were collected following the guidelines from an approved IRB protocol, using Ficoll-Paque. CD4+ memory T cells were isolated by human memory CD4 T cell enrichment kit (19157, Stemcell). Cells were cultured as a million cells per ml in RPMI media supplemented with 10% FBS and 15 units/ml of IL2. Cells were treated with 0 or 5 μM of GSK-J4 for 72 hours. Total genomic DNA was isolated using DNeasy Blood & Tissue Kit (69504, Qiagen). Bisulfite conversion of DNA was performed by EZ DNA methylation-lightning kit (D5030, Zymo research) using 1.5 μg of DNA per reaction. Nested PCR to amplify the 5' LTR CpG cluster of HIV-1 was performed as described previously [63] using 125 ng of bisulfite-treated DNA per reaction. Ion torrent A and P1 sequencing adapters and barcode sequences were incorporated to the primers used for the second round of PCR by oligo synthesis. The amplified PCR products were gel purified and loaded on the ion 540 or 520 Chip for next-generation sequencing using the Ion Chef and Ion S5 sequencing system (ThermoFisher) following manufacturer's protocol. CpG methylation was analyzed by QUMA with the default settings [95] using the 5'LTR sequence from the HBX2 strain as reference.

## Supporting information

**S1 Fig. HIV-1 reactivation in UTX knocked down E4 clones.** (A) percentage of d2EGFP positive subpopulation of cells versus (B) arithmetic mean of fluorescence intensity of d2EGFP in the population. Fold induction was calculated relative to untreated condition for each shRNA treatment.
(TIF)

**S2 Fig. UTX functions as a transcription activator of HIV-1 transcription.** ChIP assays measuring the enrichments of (A) RNAPII, (B) UTX, (C) EZH2, and (D) H3K27me3 at the Nuc-0, HIV-1 promoter, Nuc-1, and Nuc-2 regions of latent or reactivated HIV-1. Latent proviruses in E4 cells were reactivated by TNFα (10 ng/ml) for an hour. Cells were treated with ethylene glycol bis (succinimidyl succinate) (EGS) (1.5 mM) for 30 minutes and subsequently crosslinked with formaldehyde (1%) for an additional 10 minutes. ChIP assays measuring the enrichment of (E) RNAPII, (F) UTX, (G) H3K4me3, and (H) H3K27me3 along HIV-1 genome in one UTX knocked down clone. Error bars: SEM of 3 separate quantitative real-time PCRs. The same amount of rabbit IgG was used for immunoprecipitation per each treatment condition. Primers used for qPCR: HIV-1 Nuc-0: -390F, ACA CAC AAG GCT ACT TCC CTG A, and -283 R, TCT ACC TTA TCT GGC TCA ACT GGT; HIV-1 promoter: -116 F, AGC TTG CTA CAA GGG ACT GGT, and +4 R, ACC CAG TAC AGG CAA AAA GCA G; HIV-1 Nuc-1: +30 F, CTG GGA GCT CTC TGG CTA ACT A, and Nuc-1- R, TTT CAA GTC CCT GTT CGG GCG and HIV-1 Nuc-2: +283 F, GAC TGG TGA GTA CGC CAA AAA T, and +390 R, TTT CCC ATC GCG ATC TAA TTC.
(TIF)

**S3 Fig. Flow cytometry assay for latency reversal in E4 cells.** Representative FACS measuring d2EGFP expression in E4 cells are shown for cells pretreated with increasing concentrations of GSK-J4 for 24 hours and stimulated with SAHA (2 μM) or TNFα (10 ng/ml) overnight.
(TIF)

**S4 Fig. EC50 and CC50 data of GSK-J4 on HIV-1 latently infected Jurkat cell lines and primary Th17 T cell.** Cells were treated with increasing concentrations of GSK-J4 for 24 or 48

hours. Viability of cells was measured by FACs after being stained with Fixable Viability Dye eFluor 450 (#65-0863-14, ThermoFisher). Viability of cells was presented relative to cells treated with 0 nM of GSK-J4. Intracellular H3K27me3 levels of cells was measured by FACs using Tri-Methyl-Histone H3 (Lys27) antibody (#12158S, Cell Signalling) at a dilution of 1:500.
(TIF)

**S5 Fig. ChIP assays.** Enrichment of (B) RNAPII, (C) H3K4me3, and (D) H3K27me3 at the 5'LTR of HIV-1 when latent proviruses were left unstimulated or reactivated for an hour by TNFα (10 ng/ml) with or without the presence of GSK-J4 (5 μM). E4 cells were pretreated for 24 hours with GSK-J4 (5 μM) then further stimulated with or without TNFα (10 ng/ml) for an hour. Error bars: SEM of 3 separate quantitative real-time PCRs.
(TIF)

**S6 Fig. Quantification of HIV-1 reactivation by SAHA or TNFα in E4 cells expressing mock, WT H3.3 or K27M H3.3 mutants.** Cells were treated with SAHA (1 μM) or TNFα (1 ng/ml) overnight. HIV-1 reactivation was measured by FACs. Note that incorporation of WT H3.3 into HIV-1 chromatin does not predispose latently infected cells to viral reactivation.
(TIF)

**S7 Fig. Silencing kinetics of HIV-1 in Th17 cells from 4 different donors in the presence of 0 μM (vehicle) or 1 μM of GSK-J4 presented individually.** Data presented at each time point were the averages of duplicated measurements with FACs. Note that at Day 8, we did not have enough cells for collection with donor 2.
(TIF)

**S8 Fig. GSK-J4 inhibits the reactivation of latent HIV-1 by SAHA & IL15 or TCR stimulation in Th17 primary cells.** (A) Experimental design. (B) Flow cytometry measuring the reactivation of HIV-1 in Th17 cells by the combination of SAHA (1 μM) and IL15 (10 ng/ml) or Human T-Activator CD3/CD28 Dynabeads in the presence of increasing concentrations of GSK-J4. Cells were pretreated for 48 hours with GSK-J4, and then further stimulated overnight with SAHA & IL15 or Human T-Activator CD3/CD28 Dynabeads in a ratio of 25 μl of beads per 1 million cells.
(TIF)

**S9 Fig. The percentage DNA methylation at individual CpG islands at the 5'LTR CpG cluster of HIV-1 from CD4+ memory T cells induced with GSK-J4.** Note that these donors are the same as those presented in Fig 8B.
(TIF)

**S10 Fig. Loss of HIV-1 DNA methylation after removal of GSK-J4.** HIV-1 DNA methylation levels at individual CpG islands along the 5'LTR of HIV-1 before (A) and after (B) removal of GSK-J4. Experiments were performed on one HIV-1 donor with duplicates.
(TIF)

**S11 Fig. Temporary inhibition of HIV-1 reactivation and induction of HIV-1 DNA methylation by GSK-J4.** The impact of GSK-J4 on HIV-1 reactivation and DNA methylation was studied using the same experimental design as in Figs 7 and 8. Data shown is an analysis of two technical replicates from a single donor. (A) EDITs assay measuring HIV-1 reactivation in cells untreated or treated with GSK-J4 (5 μM) before and after GSK-J4 removal. Relative levels were normalized to the levels of spliced env mRNA from cells treated with T-Activator CD3/CD28 Dynabeads (presented as 100%) for each time point. Due to the limit of HIV-1 donor

sample, experiments were performed on only one HIV-1 donor with duplicates. (B) Relative expression levels of CD69 from treated CD4+ memory T cells measured by FACS using CD69 antibody. Relative levels were normalized to the levels of CD69 from cells treated with T-Activator CD3/CD28 Dynabeads (presented as 100%) for each time point. (C) Relative intracellular H3K27me3 levels of treated CD4+ memory T cells measured by FACS using a H3K27me3 antibody. Cells were stained with Alexa Fluor 647-conjugated trimethyl histone H3 (Lys27) antibody (12158, Cell Signaling) and analyzed by FACS as described previously [22]. (D) Viability of cells by PI staining. After drug treatment, cells were stained with PI at the final concentration of 5 μg/ml for 5 minutes, then analyzed by FACS. (E) The average DNA methylation (as %) of CpGs at the 5'LTR CpG cluster from CD4+ memory T cells treated with the indicated concentration of GSK-J4 before and after GSK-J4 removal. (F) The percentage DNA methylation at the CpG island 1 at the 5'LTR CpG cluster of HIV-1 from CD4+ memory T cells induced with GSK-J4 before or after GSK-J4 removal.
(TIF)

**S1 Table. One-way ANOVA, n = 5 p$<$0.005, Bonferroni posttests of HIV-1 reactivation in cells expressing different H3.3 variants.**
(XLSX)

## Acknowledgments

We thank Robert Assad and Miguel E. Quiñones-Mateu for providing us blood samples from HIV-1 infected donors. We also thank former and present members of the Karn laboratory and Nga Le for help and useful discussion. We also thank the CWRU/UH Center for AIDS Research for provision of flow cytometry services.

## Author Contributions

**Conceptualization:** Kien Nguyen, Jonathan Karn.

**Data curation:** Kien Nguyen.

**Formal analysis:** Kien Nguyen, Jonathan Karn.

**Funding acquisition:** Jonathan Karn.

**Investigation:** Kien Nguyen, Won-Kyung Cho, Jonathan Karn.

**Methodology:** Kien Nguyen, Curtis Dobrowolski, Meenakshi Shukla, Won-Kyung Cho, Benjamin Luttge.

**Supervision:** Jonathan Karn.

**Validation:** Kien Nguyen, Jonathan Karn.

**Writing – original draft:** Kien Nguyen, Jonathan Karn.

**Writing – review & editing:** Kien Nguyen, Jonathan Karn.

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
