## [Decision Letter · Decision Letter 0]

21 Jul 2021

Dear Dr Nguyen,

Thank you very much for submitting your manuscript "Inhibition of the H3K27 demethylase UTX enhances the epigenetic silencing of HIV proviruses and induces HIV-1 DNA hypermethylation but fails to permanently block HIV reactivation" for consideration at PLOS Pathogens. As with all papers reviewed by the journal, your manuscript was reviewed by members of the editorial board and by independent reviewers. In light of the reviews (below this email), we would like to invite the resubmission of a significantly-revised version that takes into account the reviewers' comments.

We cannot make any decision about publication until we have seen the revised manuscript and your response to the reviewers' comments. Your revised manuscript is also likely to be sent to reviewers for further evaluation.

Sincerely,

Bryan R. Cullen

Associate Editor

PLOS Pathogens

Thomas Hope

Section Editor

PLOS Pathogens

Kasturi Haldar

Editor-in-Chief

PLOS Pathogens

orcid.org/0000-0001-5065-158X

Michael Malim

Editor-in-Chief

PLOS Pathogens

orcid.org/0000-0002-7699-2064

Reviewer's Responses to Questions

**Part I - Summary**

Reviewer #1: Authors report on the inhibition of the H3K27me3 demethylase in an effort to epigenetically silence HIV transcription/reactivation. Although this block and lock approach seems to fail due to a transient effect, the elegant experiments increase the mechanistic understanding of both histone and DNA methylation at the transcription start site.

Reviewer #2: In this manuscript, the authors investigated the effects of UTX inhibition, an H3K27 demethylase, on HIV-1 latency and reversal from latency. They performed either shRNA experiments or a pharmalogical approach (using a compound called GSK-J4) in several cellular models of HIV-1 infection or latency (T cell lines, a primary cell model of HIV-1 latency and patients cells isolated from HIV-1+ individuals). They showed that UTX inhibition induced HIV-1 latency by increasing the histone post-translational modification H3K27me3 while decreasing H3K4me3. Overall, the manuscript is well written. Several figures could be improved to provide a clearer message. In addition, despite the high interest to further understand the molecular mechanisms underlying HIV-1 latency, several additional experiments and justifications are needed to draw the conclusions stated by the authors.

Major comments:

- Figures 1B and 1C : the authors should present their results as a fold inductions compared to the result obtained in untreated condition. Indeed, the fold induction following SAHA treatment (Fig. 1C), compared to the untreated condition, is probably much higher in shUTX(1) and shUTX(2) cell lines than in the scrambled one. Regarding this result, the following conclusion made by the authors: “UTX shRNA sequences also prevented proviruses from being reactivated by SAHA stimulation” should be modified since high levels of viral reactivation occurred in shUTX(1) and shUTX(2) cell lines following SAHA treatment.

- Figure 1B : the authors should present their data similarly in the 3 panels. In addition, it is unclear why CD3/CD28 stimulation has not been performed in 2D10 and 3C9 cells. Moreover, the lack of statistical analysis and the important error bars make the effect of UTX depletion on HIV-1 reactivation difficult to analyze (especially in the 2D10 cell line following SAHA treatment and in the 3C9 cell line following TNF-� stimulation).

- Figures 2 A-D : No negative control (such an IgG antibody) was used to assess background measurement in ChIP experiments (as well as for the other ChIP assays performed in the manuscript). In addition, since Jurkat E4 cells have 2 LTRs, is it difficult to understand how the qPCR experiments using primers amplifying Nuc-0, Pro and Nuc-1 are specific of the 5’LTR without amplifying the 3’LTR.

- Figures 2 E-F : the authors explained that the shUTX(2) was less efficient than the shUTX(1). However, the experiments were performed using the shUTX(2) cell line while the shUTX(1) cell line was used in Figure 5. To strengthen their results, the authors should repeat their experiments using both shUTX cell lines or, at least, use the same shUTX cell line through the whole manuscript.

- Figure 2B versus Figure 2F : the recruitment of UTX to the promoter region was much higher in the shScr cell line than in Jurkat E4 cells (0.1% compared to 22%). However, no explanation is provided regarding this variation. The authors should also performed ChIP assays in the shScr and shUTX(2) cell lines to assess the presence of H3K27me3 and correlate UTX depletion with H3K27me3 enrichment.

- Figure 3A : GSK-J4 (100 microM) seems to increase viral transcription in E4 and 3C9 cell lines which is in contradiction with its putative repressive effect. However, this result is not explained by the authors. A similar observation is stated by the authors (lanes 228 - 229 in shUTX cells) without any explanation.

- Figures 3B - D : The condition GSK-J4 alone should be included to evaluate the effects of this compound alone on HIV-1 latency (and not only in combination with TNFalpha) to assess if it is possible to reinforce viral latency.

- It is unclear why the authors decided to use a specific histone variant (H3.3) to introduce mutations instead of the “canonical” histone H3. Indeed, this histone variant is known to display a different chromatin environment often associated with euchromatin (reviewed by Sara Martire and Laura A. Banaszynski, Nature Reviews Molecular Cell Biology, 2020). Thus, the incorporation of the histone variant H3.3 along the HIV-1 provirus could make latently-infected cells more prone to viral reactivation.

- Figure 5D : Western blot assay shows that the level of DNMT3A was very low in the T-cell line used by the authors. It is therefore less physiologically relevant to assess the role of DNMT3A as a putative DNA methylation effector of the HIV-1 5’LTR. The authors should also evaluate the role of DNMT3A in cell lines expressing higher level of DNMT3A.

- Figure 6D : the authors performed meDIP assay at day 4 post-treatment while the highest HIV-1 silencing effect was obtained at day 8 post-treatment. Why the authors did not evaluate levels of DNA methylation at day 8 post-treatment should be clarified.

Minor comments :

- Figures 3B - D : the authors should explain why they used GAPDH and TNFA as controls.

- Figure 4B : the authors should explain why statistical analyses were performed compared to the H3.3 K27M condition and not to the H3.3 wild-type condition.

- Figure 5B : since all the cells were treated with 5-aza, the reviewer assumes that the legend (grey and purple) corresponds to shScr and ShUTX(1), respectively. The authors should confirm this.

- Cellular viability following GSK-J4 treatment should be assessed in cell lines and not only when using patient cell cultures.

**Part II – Major Issues: Key Experiments Required for Acceptance**

Reviewer #1: Major questions

1. My major question relates to the pharmacology of the GSK-J4 inhibitor. The compound is used at different concentrations in different cells and cell lines and only once its effect on the viability of the cells is measured (Figure7). Authors should present data on IC50 and CC50 values and dose response curves in the different cells and cell lines used.

2. Authors downplay their own approach because of its transient effect, and imply that the effect of didehydrocortistatin A is not due to epigenetic silencing. This is an indirect conclusion at best since no data are shown with the latter compound. In fact very few data (only on DNA methylation) are given to demonstrate the transient effect of GSK-J4 (only experiment 8?). This ought to be better elaborated. Can authors exclude that more permanent changes in the histone epigenetic environment may exist? Line 357 In a similar way, they could have tested if withdrawal of GSK-J4 reactivated HIV.

Reviewer #2: See above

**Part III – Minor Issues: Editorial and Data Presentation Modifications**

Reviewer #1: Minor remarks

1. Figure 7, show specific data on other CpG islands in Suppl Figures (now only CpG-1 is shown)

2. p21 line 420-424 If H3.3 K4M inhibits K27M reactivation why does H3.3K27me3 removal occur first ?

3. Fig 1B Why is there a difference in the effect of UTX knockdown if Tat is inactive? Explain in the discussion.

4. Fig 3A: the concentrations in the X-axis are far from clear

5. In line 57, the authors refer to block-and-lock as a strategy to eliminate the latent provirus but this strategy does not eliminate the latent provirus.

6. Line 142. Did authors also look at accumulation of H3K4 methylation at promotor in UTX KD cells since UTX also influences H3K4me3 levels? In next section they report these results with the UTX inhibitor but did they see this also in UTX KD cell lines?

7. Tet1/ Tet2 KD not so convincing effect

8. Fig 6B and C. In each experiment 4 donors are tested, are these the same donors? In Fig 6B they show the individual donors; in 6C they average the donors. It may be interesting to see if the extent of H3K27me3 accumulation correlates with HIV silencing among the different donors. If they tested the same donors, they could show the individual donor colours as well in Fig6C.

9. Line 357 In the introduction they mention that DNA methylation is a more permanent, long term effect. How does this correlate with the strong decline of methylated DNA after only 72h after cpd removal?

Reviewer #2: See above

PLOS authors have the option to publish the peer review history of their article (what does this mean?). If published, this will include your full peer review and any attached files.

Reviewer #1: No

Reviewer #2: **Yes: **Carine Van Lint
---

## [Editor Report · Decision Letter 1]

7 Oct 2021

Dear Dr Nguyen,

We are pleased to inform you that your manuscript 'Inhibition of the H3K27 demethylase UTX enhances the epigenetic silencing of HIV proviruses and induces HIV-1 DNA hypermethylation but fails to permanently block HIV reactivation' has been provisionally accepted for publication in PLOS Pathogens.

Best regards,

Bryan R. Cullen

Associate Editor

PLOS Pathogens

Thomas Hope

Section Editor

PLOS Pathogens

Kasturi Haldar

Editor-in-Chief

PLOS Pathogens

orcid.org/0000-0001-5065-158X

Michael Malim

Editor-in-Chief

PLOS Pathogens

orcid.org/0000-0002-7699-2064
---

## [Editor Report · Acceptance letter]

18 Oct 2021

Dear Dr Nguyen,

We are delighted to inform you that your manuscript, "Inhibition of the H3K27 demethylase UTX enhances the epigenetic silencing of HIV proviruses and induces HIV-1 DNA hypermethylation but fails to permanently block HIV reactivation," has been formally accepted for publication in PLOS Pathogens.

Best regards,

Kasturi Haldar

Editor-in-Chief

PLOS Pathogens

orcid.org/0000-0001-5065-158X

Michael Malim

Editor-in-Chief

PLOS Pathogens

orcid.org/0000-0002-7699-2064